# The spatial separation of processing and transport functions to the interior and periphery of the Golgi stack

Hieng Chiong Tie[1], Alexander Ludwig[1], Sara Sandin[1,2], Lei Lu[1]*

[1]School of Biological Sciences, Nanyang Technological University, Singapore, Singapore; [2]NTU Institute of Structural Biology, Nanyang Technological University, Singapore, Singapore

**Abstract** It is unclear how the two principal functions of the Golgi complex, processing and transport, are spatially organized. Studying such spatial organization by optical imaging is challenging, partially due to the dense packing of stochastically oriented Golgi stacks. Using super-resolution microscopy and markers such as Giantin, we developed a method to identify en face and side views of individual nocodazole-induced Golgi mini-stacks. Our imaging uncovered that Golgi enzymes preferentially localize to the cisternal interior, appearing as a central disk or inner-ring, whereas components of the trafficking machinery reside at the periphery of the stack, including the cisternal rim. Interestingly, conventional secretory cargos appeared at the cisternal interior during their intra-Golgi trafficking and transiently localized to the cisternal rim before exiting the Golgi. In contrast, bulky cargos were found only at the rim. Our study therefore directly demonstrates the spatial separation of processing and transport functions within the Golgi complex.
DOI: https://doi.org/10.7554/eLife.41301.001

## Introduction

The Golgi complex is one of the most important processing and sorting stations along the secretory and endocytic pathway (*Glick and Luini, 2011*; *Klumperman, 2011*; *Lu and Hong, 2014*). In mammalian cells, it consists of a network of laterally linked Golgi stacks. As the structural unit, a Golgi stack comprises 4–7 flattened cisternae and can be divided into *cis*, medial and *trans*-regions. The *trans*-Golgi region further develops into the *trans*-Golgi network (TGN). It is known that the *cis*-Golgi receives secretory cargos from the endoplasmic reticulum (ER) exit site (ERES) and ER Golgi intermediate compartment (ERGIC), while the *trans*-Golgi and TGN exchange materials with endosomes and the plasma membrane (PM). At the moment, we still don't understand how the Golgi becomes organized and works at the molecular and cellular level (*Glick and Luini, 2011*). One of the challenges in studying the Golgi is to spatiotemporally resolve residents and transiting cargos among individual cisternae of Golgi stacks, a task currently beyond the capabilities of even super-resolution and electron microscopy (EM).

It has been hypothesized that the two principal functions of the Golgi, processing and transport, are spatially organized for optimal efficiency (*Patterson et al., 2008*). However, such molecular organization across the Golgi stack has not been directly demonstrated. Previously, by utilizing nocodazole-induced Golgi mini-stacks, we developed a conventional microscopy based super-resolution method, named GLIM (Golgi localization by imaging center of fluorescence mass), to quantitatively map the axial position or localization quotient (LQ) of a Golgi protein with nanometer accuracy (*Tie et al., 2017*; *Tie et al., 2016b*). To understand the molecular organization of the Golgi mini-stack, the lateral localization, which refers to the distribution of a protein within Golgi cisternal membrane sheets, is also required. Although more structural details of the Golgi can be resolved with the

*For correspondence:
lulei@ntu.edu.sg

**Competing interests:** The authors declare that no competing interests exist.

advent of the super-resolution microscopy, it is still difficult to unambiguously interpret Golgi features due to the dense packing of stochastically oriented Golgi stacks. Here, we established a method to systematically study the lateral localization of Golgi proteins. We found that Golgi enzymes and components of trafficking machinery are spatially separated to the interior and periphery, respectively, of the Golgi stack, while secretory cargos with bulky sizes are excluded from the interior during their intra-Golgi transition.

## Results

### Giantin, GPP130 and Golgin84 localize to the cisternal rim of the Golgi mini-stack

There have been extensive evidences demonstrating that the nocodazole-induced Golgi mini-stack is a valid model of the native Golgi (*Cole et al., 1996*; *Rogalski et al., 1984*; *Trucco et al., 2004*; *Van De Moortele et al., 1993*) and we have previously discussed its advantages in studying the molecular and spatial organization of the Golgi (*Tie et al., 2017*; *Tie et al., 2016b*). Apparently, the lateral localization of a Golgi protein is best revealed by its en face and side view, when the Golgi axis is roughly orthogonal and parallel, respectively, to the image plane. We found that the orientation of a mini-stack can be identified by Golgi markers, such as Giantin, Golgin84 and GPP130. Airyscan super-resolution microscopy clearly revealed their staining patterns as rings (*Figure 1A*). Assuming cisternae of a Golgi mini-stack are round membrane disks, we reasoned that these proteins must localize to the rim of their corresponding cisternae and their ring appearances must correspond to en face or oblique views (hereafter en face views) (*Figure 1B*). As expected for the orthogonal section of a ring (*Figure 1B*), side view images of Giantin, Golgin84 and GPP130 displayed a double-punctum, the connecting line of which is roughly orthogonal to the Golgi-axis (*Figure 1C*). To describe the localization pattern of a population of mini-stacks, we developed a method to average multiple en face view images of Golgi mini-stacks, by applying size and intensity normalization followed by alignment according to their centers of fluorescence mass (see Materials and methods). En face averaged Giantin, Golgin84 and GPP130 demonstrated their lateral localization patterns as concentric circular rings of difference sizes (*Figure 1D–F*). To substantiate our light microscopic data, we imaged APEX2-fused GPP130 by EM using native NRK cells that were not subjected to nocodazole treatment (*Figure 1G*; *Figure 1—figure supplement 1A,B*). Out of 57 Golgi stacks that we randomly imaged from 25 cells, 68% demonstrated a predominant cisternal rim localization in side or en face views (*Figure 1—figure supplement 1C*), supporting the ring staining pattern observed. The rim localization of Giantin was also corroborated in a previous immuno-EM study, furthring supporting our data (*Koreishi et al., 2013*).

Among all Golgi markers, we observed that Giantin had the largest ring diameters—950 ± 10 nm (mean ± SEM, same for the rest; n = 336) (*Figure 1H*). It is known that the epitope of our antibody is at the N-terminus while Giantin anchors onto the Golgi membrane via its extreme C-terminal transmembrane domain (*Linstedt et al., 1995*). A fully extended Giantin molecule is predicted to reach 450 nm (*Munro, 2011*). Hence, it is possible that the large diameter of Giantin-ring can be due to Giantin's extended structure instead of the physical dimension of Giantin-positive cisternae. However, we think this is not the case due to our following observations. First, we raised an antibody against its C-terminal cytosolic region and ring-patterns resulted from N- and C-terminal antibodies colocalized very well (*Figure 1I*). Quantitative analysis revealed that the mean diameter of the C-terminus ring is ~50 nm smaller than that of the N-terminus one (*Figure 1H*; *Figure 1—figure supplement 2A*), far less than the value predicted for the fully extended molecule, which is 900 nm. Second, the C-terminal 129 amino acid fragment of Giantin (mScarlet-Giantin-C129), which has a LQ similar to native Giantin (*Table 1*), displayed almost the same ring-pattern as the N-terminal antibody (hereafter Giantin antibody unless indicated otherwise) (*Figure 1J*). Third, similarly, the N- and C-termini of other Golgins such as GM130 and GCC185 also showed overlapping ring-patterns (*Figure 1—figure supplement 2B,C*). In summary, although individual Golgins might adopt long filamentous conformation (*Munro, 2011*), ensemble-averaged Golgins, as visualized in bulk by light microscopy, appear to have a closely adjacent N- and C-termini (*Cheung et al., 2015*). Therefore, the ring-pattern staining of Giantin should closely represent the cisternal rim.

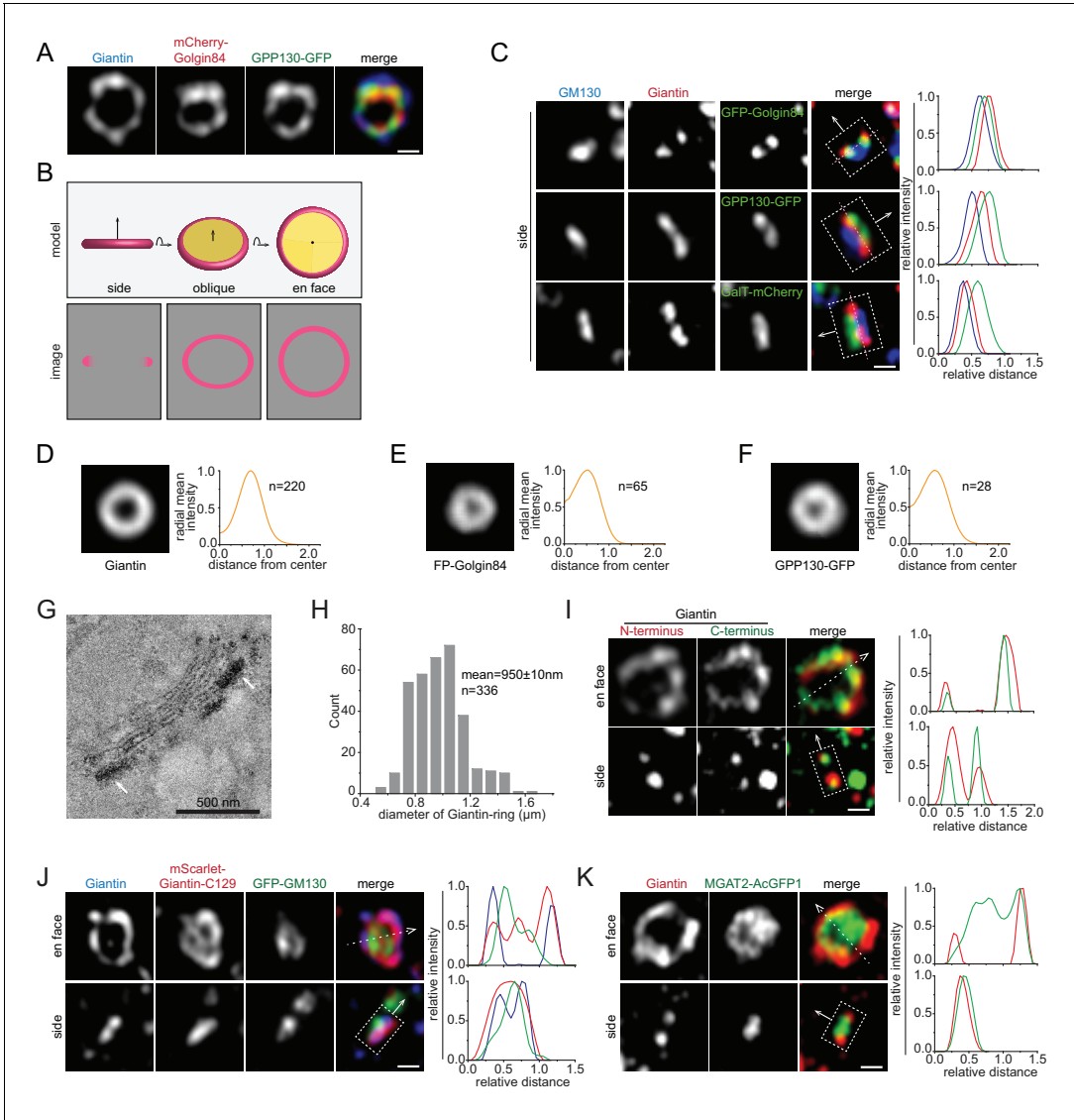

**Figure 1.** Identifying the en face and side view of the Golgi mini-stack. All cells are nocodazole-treated HeLa cells and all images are super-resolution images unless specified otherwise. By default, tagged-proteins were transiently transfected while non-tagged proteins were native and stained by their antibodies. (A) The staining patterns of Giantin, Golgin84 and GPP130 appear as concentric rings. (B) The schematic representation of different orientation views (en face, oblique and side) of a Golgi cisterna and the corresponding expected images of a rim-localized protein (colored as pink). (C) The double-punctum appearances of Giantin, Golgin84 and GPP130 indicate side views of Golgi mini-stacks. In each merge, the intensity profile is generated along a thick line, represented by a dotted box, with the direction indicated by the arrow (the same scheme is used throughout this study). The dotted box schematically marked the start, end and width of the line. The direction arrow roughly follows the *cis*-to-*trans* Golgi axis using the *cis*-most (GM130 in this case) and *trans*-most markers in each panel. Dotted pink lines connecting double-punctum are almost orthogonal to the *cis*-to-*trans* Golgi axis. The intensity plot is normalized and color-coded as the corresponding merge image. (D–F) En face averaged images of Giantin, fluorescence protein (FP)-Golgin84 and GPP130-GFP. The corresponding radial mean intensity profile is shown at the right with distance from the center of fluorescence mass (normalized to the radius of Giantin) as the x-axis and radial mean intensity (normalized) as the y-axis. Both GFP and mCherry-tagged Golgin84 images were used for FP-Golgin84. n, the number of averaged Golgi mini-stacks. (G) GPP130 mostly localizes to the cisternal rim (arrows) of the native Golgi by EM. NRK cells transiently expressing GPP130-APEX2-GFP were subjected to APEX2-catalyzed reaction followed by EM. Note that cells were not subjected to nocodazole treatment. The EM thin section image displays the side view of a Golgi mini-stack. The electron density indicates the localization of GPP130 (arrows). (H) The histogram showing the distribution of diameters of Giantin-rings. (I, J) Giantin N and C-terminus colocalize at the cisternal rim. In (I), cells were co-stained using Giantin antibodies raised against its N and C-terminus. In (J), Giantin N-terminus was stained by an antibody and its C-terminus was revealed by exogenously expressed mScarlet-Giantin-C129. In the en face view, dotted arrow represents the line used to generate the line intensity profile (width = 1 pixel), while in the side view, the dotted box that is in the direction of the arrow and parallel to the Golgi cisterna represents the line for intensity profile. (K) The interior localization of MGAT2 within the Giantin-ring. Line intensity profiles of the en face and side views are acquired as those in (I) and (C) respectively. Scale bar, 500 nm.

*Figure 1 continued on next page*

*Figure 1 continued*

DOI: https://doi.org/10.7554/eLife.41301.002

The following figure supplements are available for figure 1:

**Figure supplement 1.** GPP130 displays rim-localization in a majority of native Golgi stacks by EM.
DOI: https://doi.org/10.7554/eLife.41301.003
**Figure supplement 2.** The N and C-terminus of Giantin, GCC185 and GM130 coincide on the Golgi mini-stack.
DOI: https://doi.org/10.7554/eLife.41301.004

## Identifying en face and side views of Golgi mini-stacks

By assessing the super-resolution staining patterns of Giantin, GPP130 or Golgin84, we can conveniently identify en face and side view oriented Golgi mini-stacks, images of which should appear as a ring and double-punctum, respectively. It was discovered that some Golgi residents, such as MGAT2, localized to the interior of Giantin-rings (*Figure 1K*). Consistent with this interpretation, side views of MGAT2 appeared as a short bar connecting the Giantin double-punctum (*Figure 1K*). Under the EM, MGAT2-APEX2-GFP preferentially localized to the cisternal interior (next section). Therefore, there are at least two types of lateral localizations: rim and interior, as represented by Giantin and MGAT2.

## Golgi trafficking components mainly localize to the periphery of a Golgi mini-stack

We systematically examined the lateral localization of Golgi residents using their en face and side views. Two types of residents were studied in this work — components of trafficking machinery, including those involved in the structure and organization of the Golgi, and enzymes involved in the post-translational modifications, particularly glycosyltransferases. Due to the lack of reagents to detect endogenous proteins, many residents were detected by the overexpression of their tagged fusions (*Table 1*). Caution must be taken in the interpretation of our data as it has been documented that overexpression can change both the axial and lateral localization of Golgi residents (*Cosson et al., 2005*). We discovered that the lateral localization of trafficking machinery components shares common features according to their LQs.

### ERES, ERGIC and cis-Golgi proteins (LQ <0)

COPII coat subunits, including Sec13 and Sec23a, COPI coat subunits, including β and γ-COP, KDEL receptor, GS27, ERGIC53, Arf4 and Arf5, displayed lumps or puncta around Giantin-rings in en face views and at one side of Giantin-double-punctum in side views (*Figure 2A–D*; *Figure 2—figure supplement 1A–E*).

### cis-Golgi proteins ($0 \leq LQ < 0.25$)

GM130, GRASP55, GRASP65 and Rab1a mainly appeared as a central disk and bar in en face and side views, respectively (*Figure 2E–I*; *Figure 2—figure supplement 1F,G*). When they appeared as rings in en face views, there were usually some interior tubular or sheet connections (*Figure 2E,H*; *Figure 2—figure supplement 1F*). Both observations suggest that these proteins probably localize throughout *cis*-cisternae.

### Medial and trans-Golgi proteins ($0.25 \leq LQ < 1.0$)

ACBD3, Golgin84, Giantin, GS15, GS28, Sec34, GPP130 and GCC185, all displayed ring-pattern localizations (*Figure 1A,C*; *Figure 3A–F*; *Figure 3—figure supplement 1A–D*), suggesting that they mainly localize to the rim of their corresponding cisternae and are mostly absent from the cisternal interior. Arf1, whose LQ is 0.75, is an exception here. Although its en face view demonstrated that it is in the cisternal interior, side view images uncovered that there were two pools: a *cis*/medial and a *trans*-Golgi/TGN pool, with a much reduced presence in between (*Figure 3G,H*). This observation is consistent with the notion that Arf1 functions in the *cis*-Golgi and TGN for the assembly of the COPI and clathrin coat, respectively (*Gillingham and Munro, 2007*).

**Table 1.** List of LQs used in this study.

Please see *Table 1*-table supplement 1 for official full names of glycosylation enzymes.

| Name | LQ | N | SEM |
|---|---|---|---|
| Myc-Sec13 | −0.96 | 39 | 0.09 |
| β-COP[$] | −0.70 | 74 | 0.11 |
| Arf4-GFP | −0.61 | 51 | 0.07 |
| Sec23a-mCherry | −0.58 | 121 | 0.06 |
| Arf5-GFP | −0.46 | 42 | 0.06 |
| GS27[*,$] | −0.22 | 101 | 0.03 |
| γ-COP[$] | −0.17 | 106 | 0.07 |
| GFP-ERGIC53[*] | −0.16 | 198 | 0.02 |
| KDEL receptor[*, $] | −0.11 | 130 | 0.03 |
| GFP-GM130[*] | −0.05 | 93 | 0.04 |
| GM130[*, $, #] | 0.00 | - | - |
| GRASP65-GFP | 0.02 | 198 | 0.01 |
| GRASP55-GFP | 0.07 | 140 | 0.02 |
| GFP-Rab1a | 0.21 | 154 | 0.03 |
| ManII-SBP-GFP | 0.23 | 53 | 0.05 |
| GFP-ACBD3 | 0.25 | 132 | 0.03 |
| GFP-Golgin84[*] | 0.26 | 108 | 0.03 |
| Man1B1-Myc | 0.42 | 88 | 0.05 |
| β3GalT6-Myc | 0.47 | 97 | 0.03 |
| MGAT4B-AcGFP1 | 0.50 | 23 | 0.04 |
| β4GalT7-Myc | 0.52 | 110 | 0.04 |
| MGAT2-Myc | 0.53 | 136 | 0.04 |
| GS28[*] | 0.53 | 125 | 0.08 |
| MGAT2-AcGFP1 | 0.56 | 110 | 0.04 |
| Giantin[$] | 0.57 | 103 | 0.05 |
| TPST2-GFP[*] | 0.64 | 154 | 0.02 |
| POMGNT1-Myc | 0.67 | 87 | 0.04 |
| MGAT1-Myc | 0.70 | 141 | 0.02 |
| GPP130-APEX2-GFP | 0.71 | 100 | 0.03 |
| Myc-Sec34 | 0.71 | 27 | 0.12 |
| β4GalT3-Myc | 0.74 | 149 | 0.02 |
| Arf1-GFP | 0.75 | 87 | 0.03 |
| TPST1-GFP[*] | 0.76 | 111 | 0.04 |
| ST6Gal1-Myc | 0.76 | 154 | 0.03 |
| mScarlet-Giantin-C129 | 0.80 | 161 | 0.01 |
| GS15[$] | 0.83 | 150 | 0.03 |
| GPP130-GFP[*] | 0.84 | 168 | 0.02 |
| SLC35C1-Myc | 0.84 | 85 | 0.04 |
| ST6Gal1-AcGFP1 | 0.85 | 138 | 0.02 |
| GALNT2[$] | 0.86 | 107 | 0.03 |
| GFP-GCC185 | 0.94 | 122 | 0.05 |
| GALNT1[$] | 0.97 | 90 | 0.02 |
| GalT-mCherry[*,#] | 1.00 | - | - |

*Table 1 continued on next page*

Table 1 continued

| Name | LQ | N | SEM |
|---|---|---|---|
| GFP-Rab6[*] | 1.04 | 262 | 0.04 |
| Arl1[*, $] | 1.20 | 26 | 0.05 |
| Vti1a[*, $] | 1.26 | 162 | 0.02 |
| GFP-GGA1 | 1.30 | 33 | 0.12 |
| Golgin245[*, $] | 1.42 | 126 | 0.05 |
| GFP-Golgin97[*] | 1.45 | 161 | 0.03 |
| CI-M6PR[*, $] | 1.46 | 42 | 0.24 |
| Syntaxin6[*, $] | 1.56 | 84 | 0.11 |
| Vamp4-GFP[*] | 1.57 | 157 | 0.04 |
| Furin[*, $] | 1.62 | 43 | 0.11 |
| CLCB[$] | 1.65 | 37 | 0.26 |
| GGA2[*, $] | 1.96 | 33 | 0.23 |

[*], previously published data (**Tie et al., 2016b**);

$, endogenous protein.

#, LQs of GM130 and GalT-mCherry are defined as 0.00 and 1.00 (**Tie et al., 2016b**).

DOI: https://doi.org/10.7554/eLife.41301.005

## *trans*-Golgi and TGN proteins (LQ ≥1.0)

There are two types of localization patterns at the *trans*-side of Giantin-rings. The distribution of Vamp4, Golgin97, Vti1a, Syntaxin6, Rab6, Arl1 and Golgin245 was relatively compact (*Figure 3I,J*; *Figure 3—figure supplement 2A–G*). In contrast, GGA1, GGA2, clathrin light chain B (CLCB), CI-M6PR and Furin showed punctate or tubular profiles (*Figure 3I,K,L*; *Figure 3—figure supplement 2H–L*). Although all are TGN proteins, most of them did not exhibit appreciable colocalization. For example, Vamp4 did not show a significant overlap with CI-M6PR, GGA2, Furin, or Vti1a (*Figure 3L*; *Figure 3—figure supplement 2K–M*). However, CLCB was found to decorate punctate and tubular profiles of both Vamp4 (*Figure 3I*) and Furin (*Figure 3—figure supplement 2H*) outside the stacked cisternal membrane, in agreement with 3D EM-tomography of the TGN (*Ladinsky et al., 1999*) and the role of clathrin coat in transporting these cargos to the endolysosome (*Peden et al., 2001*; *Teuchert et al., 1999*). Our data are also consistent with the notion that the TGN comprises domains of distinct molecular compositions (*Brown et al., 2011*; *Derby et al., 2004*).

In summary, our extensive super-resolution imaging data suggest that Golgi trafficking components mainly localize to the entire *cis*-cisternae, rim of medial and *trans*-cisternae and punctate or tubular profiles at non-stacked regions, which include the ERES, ERGIC and TGN.

## Glycosylation enzymes reside at the interior of a Golgi stack

We studied components of Golgi post-translational modification machinery (*Table 1*; *Supplementary file 1*), including a GDP-fucose transporter, SLC35C1 (*Lübke et al., 2001*), and more than a dozen enzymes involved in N-glycosylation (Man1B1, MGAT1, ManII, MGAT2, GalT, SialT and MGAT4B), O-glycosylation (GALNT1, GALNT2 and POMGNT1), poly-N-acetyllactosamine synthesis (β4GalT3), glycosaminoglycan synthesis (β3GalT6 and β4GalT7) and sulfation (TPST1 and 2). Interestingly, their LQs were found to be in the range from 0.23 to 1.0 (*Table 1*), suggesting that Golgi enzymes mainly localize to the medial and *trans*-region of the Golgi, but not to the *cis*-Golgi and TGN. This observation is consistent with previous EM studies. For example, in plant cells, polysaccharides were mainly detected in the medial and *trans*-Golgi cisternae (*Zhang and Staehelin, 1992*). Similarly, in mammalian cells, the N-glycan modifying enzymes ManI, ManII and MGAT1 have been mapped to the *medial* and *trans*-region of the Golgi stack (*Dunphy et al., 1985*; *Nilsson et al., 1993*; *Rabouille et al., 1995*; *Velasco et al., 1993*). However, in contrast to our quantitative results, previous EM work has assigned GalT (*Nilsson et al., 1993*; *Rabouille et al., 1995*; *Roth and Berger, 1982*) and SialT (*Rabouille et al., 1995*; *Roth et al., 1985*) to the TGN in addition

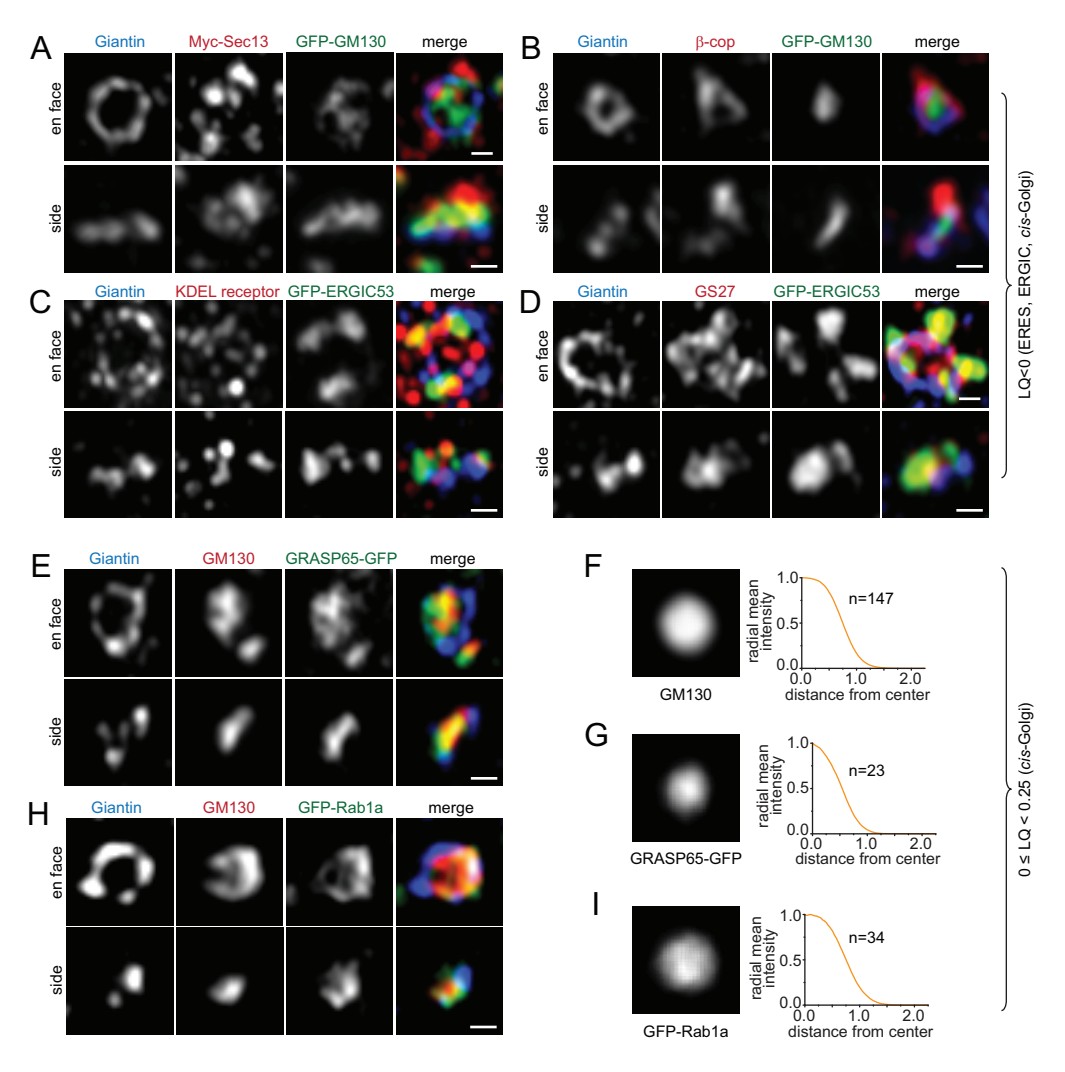

**Figure 2.** Components of the ERES, ERGIC and *cis*-Golgi transport machinery mainly localize to the periphery of the Golgi mini-stack. (**A–D, E and H**) Typical en face and side view images of Golgi transport machinery components. (**A–D**) ERES, ERGIC and *cis*-Golgi proteins (LQ <0). (**E and H**) *cis*-Golgi proteins (0 ≤ LQ < 0.25). (**F–I**) En face averaged images and radial mean intensity profiles corresponding to (**E**) and (**H**). n, the number of averaged Golgi mini-stack images. Scale bar, 500 nm.

DOI: https://doi.org/10.7554/eLife.41301.006

The following figure supplement is available for figure 2:

**Figure supplement 1.** Typical en face and side view images of Golgi transport machinery components.
DOI: https://doi.org/10.7554/eLife.41301.007

to the *trans*-Golgi. Sub-Golgi localizations are not always consistently reported, which is likely due to two reasons. First, the *cis*, medial, *trans*-region and TGN are not rigorously defined and the assignment of Golgi regions can be subjective. Second, it has been documented that the sub-Golgi localization of enzymes can be cell-type dependent (*Velasco et al., 1993*).

In contrast to trafficking components, our Golgi enzymes and SLC35C1 localized within Giantin-rings as a central disk in en face views (*Figure 1K*; *Figure 4A–D*; *Figure 4—figure supplement 1A–T*), except Man1B1, ManII, MGAT4B and TPST2, which mostly appear as an inner-ring concentric to the corresponding Giantin-ring (*Figure 4E,F*; *Figure 4—figure supplement 2A–F*). The disk and ring patterns were more obviously revealed after en face averaging (*Figure 4B,D,F*; *Figure 4—figure supplement 1B,D,F,H,J,L,N,P,R,T*; *Figure 4—figure supplement 2B,D,F*). Since MGAT2-Myc and MGAT4B-AcGFP1 had almost the same LQs as Giantin (mean values: 0.53 and 0.50 vs 0.57

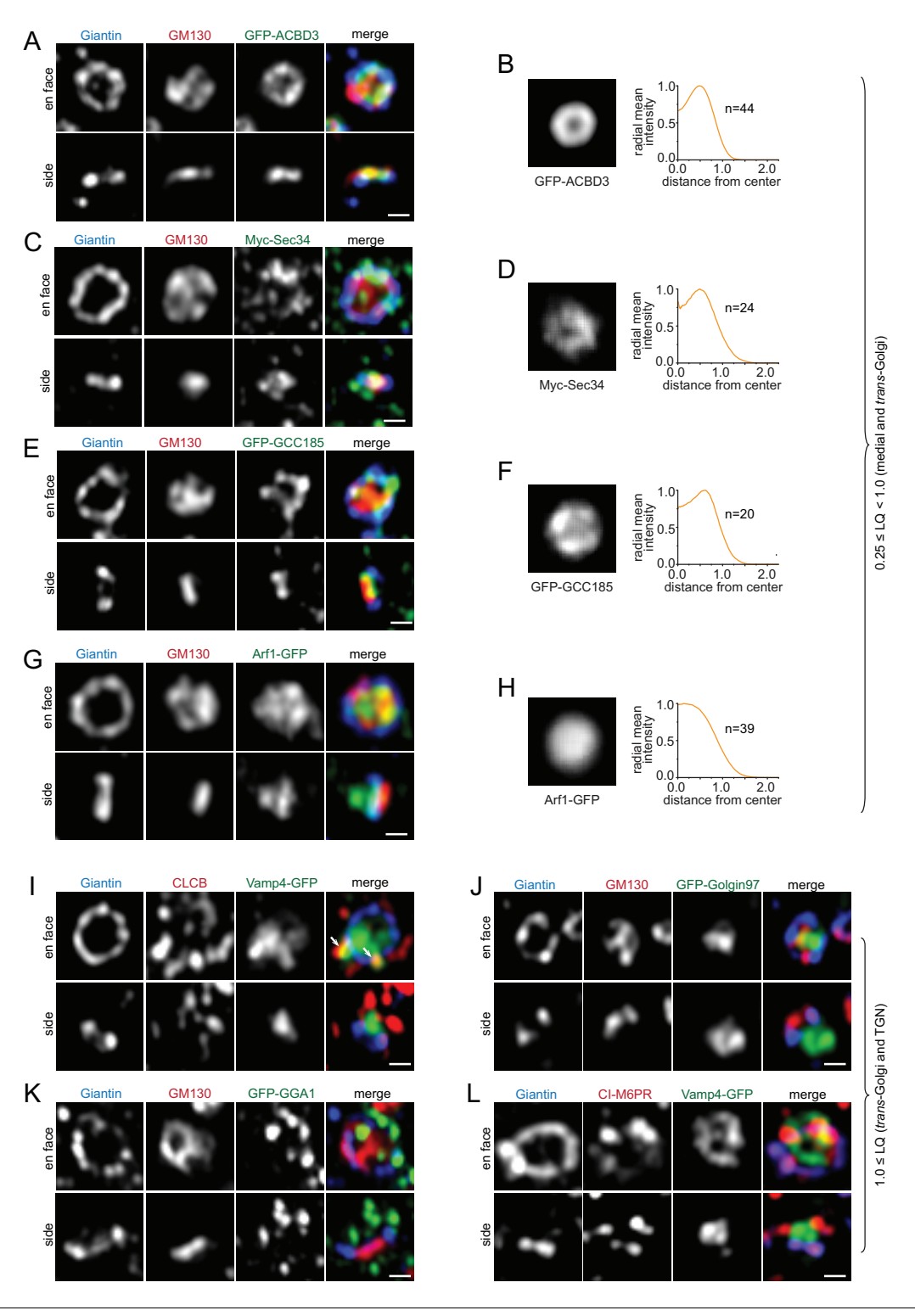

**Figure 3.** Components of the medial, *trans*-Golgi and TGN transport machinery mainly localize to the periphery of the Golgi mini-stack. (**A–H**) Medial and *trans*-Golgi proteins ($0.25 \leq LQ < 1.0$), except Arf1, localize to the cisternal rim. En face and side view images are shown. Corresponding en face averaged images and radial mean intensity profiles are shown in (**B, D, F and H**). n, the number of averaged Golgi mini-stack images. (**I–L**) *trans*-Golgi and TGN proteins ($LQ \geq 1.0$) appear compact or scattered at one end of the mini-stack. Arrows in (**I**) indicate colocalization between CLCB and Vamp4-GFP. Scale bar, 500 nm.

*Figure 3 continued on next page*

*Figure 3 continued*

DOI: https://doi.org/10.7554/eLife.41301.008

The following figure supplements are available for figure 3:

**Figure supplement 1.** En face and side view images of the medial and *trans*-Golgi SNAREs, including GS15 and GS28, showed their rim localization (**A and C**).

DOI: https://doi.org/10.7554/eLife.41301.009

**Figure supplement 2.** The lateral localization of components of the *trans*-Golgi and TGN transport machinery in the Golgi mini-stack.

DOI: https://doi.org/10.7554/eLife.41301.010

respectively) (*Table 1*), a significant amount of these proteins are expected to reside in the same cisternae. The lateral distribution pattern of MGAT2 and MGAT4B suggests that they should mainly localize to the interior of cisternae as a central disk and inner-ring, respectively, within the Giantin-rim in the same cisternae (*Figure 4G*). Enzymes, such as β4GalT3 and ST6Gal1, which have similar LQs (*Table 1*), were observed to localize to shared and distinct domains within Giantin-rings (*Figure 4H*).

To substantiate our light microscopic data, we examined the localization of MGAT2-APEX2-GFP in the native Golgi by EM. 93% (n = 58) of Golgi stacks showed an enrichment of MGAT2 in the cisternal interior (*Figure 4I*; *Figure 4—figure supplement 3A–C*), which is in contrast to the staining pattern observed for GPP130 (*Figure 1G*). Noticeably, APEX2-generated electron density was also found in vesicles and budding profiles at the rim (arrow heads in *Figure 4I*). However, we did not find MGAT2-AcGFP1 (*Figures 1K* and *4C*) or MGAT2-APEX2-GFP (*Figure 4—figure supplement 1U*) signal outside Giantin-rings by fluorescence imaging of Golgi mini-stacks. Although the identity and destiny of these vesicles are currently unknown, our observations suggest that Golgi enzymes might be depleted from the rim either by retrieval to the interior or by sorting into membrane carriers. Together, our data demonstrate that Golgi enzymes mainly localize to the interior of medial and *trans*-cisternae as a concentric disk or inner-ring, while trafficking machinery components exhibit rim localization.

## A quantitative molecular map of the Golgi mini-stack

To quantitatively describe the overall lateral distribution of Golgi proteins, we assume that a Golgi protein has a radial symmetry localization around the Golgi axis as a concentric disk or ring. The normalized radius of the ring or disk can be measured using the radial mean intensity profile of en face averaged images (see Materials and methods). A plot of the normalized radius versus LQ quantitatively summarizes our morphological observations of ring and disk distribution of various Golgi residents (*Figure 4J*). While medial and *trans*-Golgi trafficking machinery components are at the cisternal rim, Golgi enzymes all localize to the interior with Man1B1, ManII, MGAT4B and TPST2 appearing as concentric inner-rings and the rest as central disks. Interestingly, it also reveals that *cis*-cisternae have smaller diameters than medial ones, consistent with many EM thin-section or tomographic 3D images (*Bykov et al., 2017*; *Engel et al., 2015*; *Staehelin and Kang, 2008*), though the biological significance of which remains to be further investigated.

## Imaging the organization of the native Golgi complex

Having studied in detail the organization of Golgi mini-stacks, we attempted to resolve the organization of the native Golgi complex by the super-resolution microscopy. Giantin and Golgi enzymes were used to mark the rim and interior of stacked cisternae, respectively. In the less dense region, Giantin and GPP130 staining appeared as distinctive ring- or loop-patterns, with β4GalT3 and GM130 filling the interior (*Figure 4K,L*), similar to the nocodazole-induced mini-stack. β4GalT3 and GM130 positive membrane sheets likely correspond to stacked Golgi cisternae. In most cases, Giantin and GPP130 positive curvy lines did not correspond to side views or cross-sections of Golgi stacks. Instead, they corresponded to the rim of cisternae in oblique or en face views (arrows in *Figure 4L*). In the more densely packed region, cisternae appeared to pile on top of each other, a configuration that requires much higher z-axis resolution to be resolved. Nonetheless, we

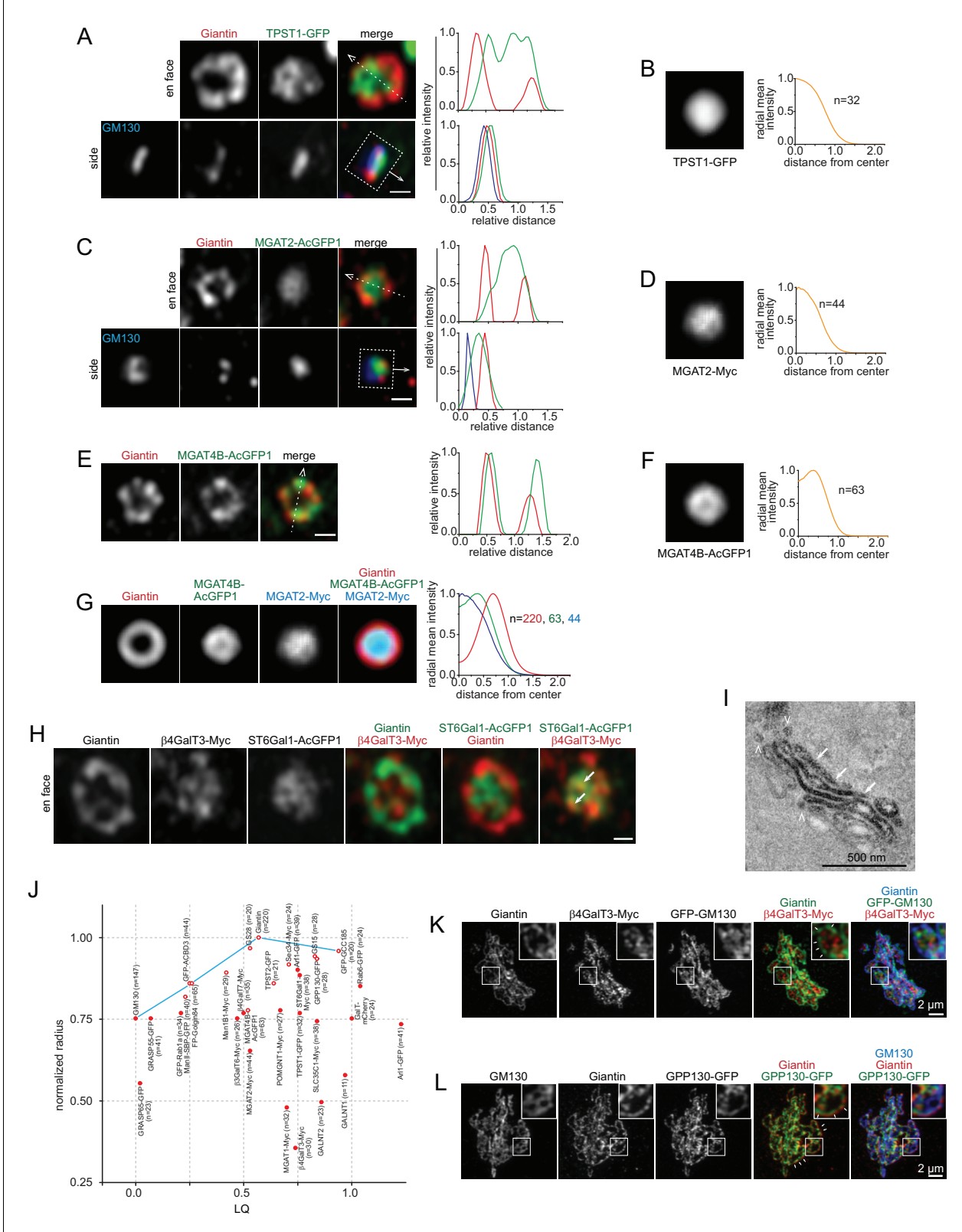

**Figure 4.** Golgi enzymes primarily localize to the interior of medial and *trans*-Golgi cisternae. (**A, C and E**) En face view images of Golgi enzymes. Side view images are also shown in (**A**) and (**C**). Dotted arrows and boxes and line intensity profiles are used or acquired as in *Figure 1K*. (**B, D and F**) Corresponding en face averaged images and radial mean intensity profiles. n, the number of averaged Golgi mini-stack images. (**G**) The merge of en face averaged images of Giantin, MGAT4B and MGAT2 and the corresponding radial mean intensity profile. n, the number of averaged Golgi mini-

*Figure 4 continued on next page*

*Figure 4 continued*

stack images. (**H**) β4GalT3 and ST6Gal1 can localize to shared (arrows) and distinct domains within the cisternal interior. (**I**) MGAT2 localizes to the cisternal interior of the native Golgi by EM. NRK cells transiently expressing MGAT2-APEX2-GFP were subjected to APEX2-catalyzed reaction followed by EM. Note that cells were not subjected to nocodazole treatment. The thin section EM image displays the side view of a Golgi stack. MGAT2-APEX2 positive cisternal interior and budding profiles are indicated by arrows and arrow heads, respectively. (**J**) A quantitative molecular map of the Golgi mini-stack. The normalized radius of a Golgi protein is plotted against its corresponding LQ (*Table 1*). Red open and closed circle denote ring and disk lateral localization pattern, respectively. n, the number of Golgi mini-stacks used to calculate normalized radius. (**K,L**) Identifying the rim and interior of native Golgi cisternae. Cells were not treated with nocodazole. In (**K**), the cisternal rim (arrows) and interior are labeled by Giantin and β4GalT3, respectively. In (**L**), Giantin and GPP130 positive curvy lines (arrows) represent cisternal rim and do not correspond to side views or cross sections of Golgi stacks. The boxed region in each image is enlarged in the upper right corner. Scale bars represent 500 nm unless specified otherwise.

DOI: https://doi.org/10.7554/eLife.41301.011

The following figure supplements are available for figure 4:

**Figure supplement 1.** Golgi enzymes that primarily display central disk localization at the interior of medial and *trans*-Golgi cisternae.

DOI: https://doi.org/10.7554/eLife.41301.012

**Figure supplement 2.** Golgi enzymes, Man1B1, ManII and TPST2, display ring-pattern localization.

DOI: https://doi.org/10.7554/eLife.41301.013

**Figure supplement 3.** MGAT2 mainly localizes to the cisternal interior in native Golgi stacks by EM.

DOI: https://doi.org/10.7554/eLife.41301.014

demonstrated that, aided with suitable markers, it is possible to identify the cisternal rim and interior of the native Golgi complex by light microscope.

## The lateral localization of secretory cargos during their intra-Golgi trafficking

To study the lateral localization of secretory cargos during their intra-Golgi trafficking, the retention using selective hooks (RUSH) system was adopted to synchronously release secretory cargos (*Boncompain et al., 2012*). The RUSH reporter CD59, a GPI-anchored protein, was first detected in the interior of *cis*-Golgi cisternae after 10 min of chase (*Figure 5A,B*). During its transition through the Golgi mini-stack, as evidenced in its LQ versus time plot (*Figure 5B*), CD59 remained in the interior (*Figure 5A*), although its total intensity in Golgi mini-stacks initially increased and subsequently decreased due to the export toward the PM. At the later stage of the chase, there were CD59 positive puncta and tubular profiles outside Giantin-rings, which were likely Golgi-derived exocytic transport carriers (*Figure 5A*, arrows in 60 min). Similarly, in live-cell super-resolution imaging, RUSH reporter mCherry-GPI started to appear in the interior of the Golgin84-ring 6 min after chase; it remained there for >30 min before disappearing due to post-Golgi exocytic trafficking (*Figure 5C*; *Figure 5—video 1*). Transmembrane RUSH reporters, E-cadherin, VSVG and CD8a-Furin, and a soluble secretory reporter, signal peptide fused GFP, followed similar lateral localization pattern during their intra-Golgi trafficking (*Figure 5—figure supplement 1A–D*). Collectively, our data demonstrated that conventional secretory cargos partition to the interior of the cisternae during their Golgi transition.

The secretory cargo wave does not seem to grossly affect the interior distribution of Golgi enzymes, as evidenced by ST6Gal1 (*Figure 5—figure supplement 2*). By image quantification, >85% of ST6Gal1-moxGFP was found to remain in the interior during the Golgi transition of synchronized VSVG, although a small fluctuation (<4%) was noticed (*Figure 5—figure supplement 2A–C*). Our finding is different from a previous EM study, in which the shift of Golgi enzymes from the rim to the interior was observed under a traffic wave (*Kweon et al., 2004*). A more systematic investigation is required to resolve this discrepancy.

## Bulky size prevents the localization of secretory cargos at the cisternal interior

Based on EM data, Rothman lab previously proposed that large secretory protein aggregates are segregated to the cisternal rim (*Lavieu et al., 2013*). To investigate if bulky cargos partition to the rim, we imaged the RUSH reporter GFP-collagenX, a soluble secretory protein that tends to form oligomers (*Kwan et al., 1991*), by Airyscan super-resolution microscopy. We observed that Golgi-transiting GFP-collagenX appeared either diffuse or punctate (*Figure 5D*). Assuming that Golgi-

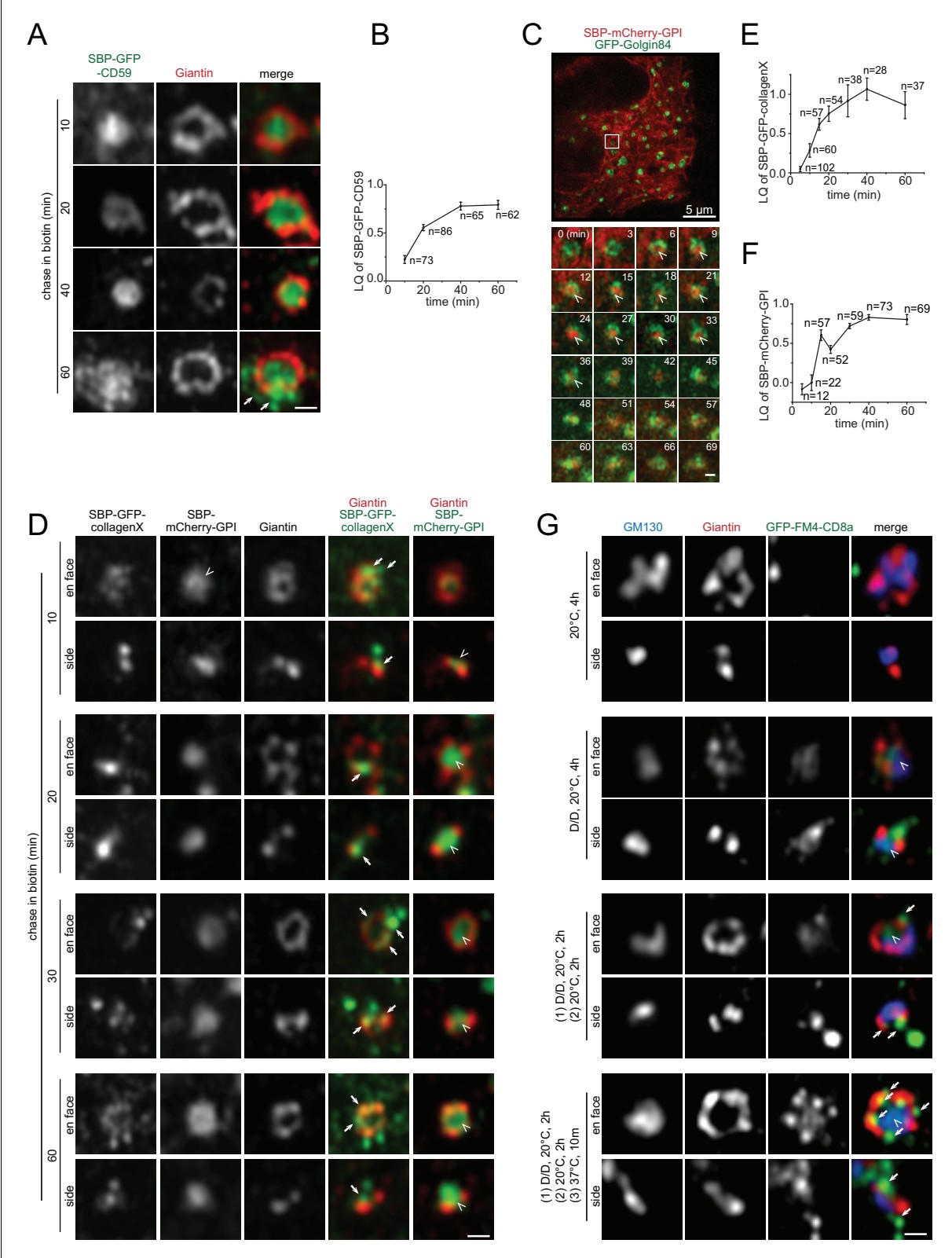

**Figure 5.** Conventional or small size secretory cargos can localize to the cisternal interior while bulky ones are restricted to the rim during their intra-Golgi transport. (A,B) CD59 localizes to the cisternal interior during its intra-Golgi transport. Cells transiently expressing RUSH reporter, SBP-GFP-CD59, were chased in the presence of biotin for indicated time (A). Arrows indicate budding membrane carriers. In (B), the LQ vs time plot measured from the same experiment demonstrated the intra-Golgi transport of CD59. (C) Live-cell imaging showing the interior localization of mCherry-GPI during its

*Figure 5 continued on next page*

*Figure 5 continued*

transition through the Golgi mini-stack. Cells transiently co-expressing RUSH reporter, SBP-mCherry-GPI, and GFP-Golgin84 were chased in biotin and imaged live under Airyscan super-resolution microscopy. The boxed region in the upper image, which was acquired before the chase, is selected to show the time series. Arrow heads indicate the interior localization. See also *Figure 5—video 1*. (D–F) The partition of collagenX and mCherry-GPI to the cisternal rim and interior respectively during their intra-Golgi transport. Cells transiently co-expressing RUSH cargos, SBP-GFP-collagenX and SBP-mCherry-GPI were chased as in (A). Arrows and arrow heads indicate the cisternal rim and interior localization respectively. The intra-Golgi transport of collagenX and mCherry-GPI was demonstrated by LQ vs time plots measured from the same experiments in (E) and (F). Error bar, mean ± SEM. n, the number of Golgi mini-stacks used for the calculation. (G) GFP-FM4-CD8a partitions to the cisternal rim upon aggregation. NRK cells transiently expressing GFP-FM4-CD8a were subjected to a combination of D/D solubilizer treatment and wash out at either 20°C or 37°C, as indicated. First set of images is the negative control showing that aggregated GFP-FM4-CD8a was not exported from the ER. Aggregated GFP-FM4-CD8a partitioned to the rim (arrows), while non-aggregated form was still interior-localized (arrow heads). Scale bars represent 500 nm unless specified otherwise.
DOI: https://doi.org/10.7554/eLife.41301.015

The following video and figure supplements are available for figure 5:

**Figure supplement 1.** Conventional or small size secretory cargos can localize to the cisternal interior during their intra-Golgi transport.
DOI: https://doi.org/10.7554/eLife.41301.016
**Figure supplement 2.** ST6Gal1 mainly localizes to the cisternal interior under VSVG traffic wave.
DOI: https://doi.org/10.7554/eLife.41301.017
**Figure supplement 3.** Bulky secretory cargos are restricted to the cisternal rim during their intra-Golgi transport.
DOI: https://doi.org/10.7554/eLife.41301.018
**Figure 5—video 1.** Live-cell imaging showing the interior localization of mCherry-GPI during its transition through the Golgi mini-stack.
DOI: https://doi.org/10.7554/eLife.41301.019

localized GFP-collagenX puncta were single multimeric aggregates, using GFP-tagged nucleoporin Nup133 as an in vivo GFP fluorescence standard, we estimated that Golgi-transiting GFP-collagenX puncta had 190 ± 20 copies (n = 77) (*Figure 5—figure supplement 3A,B*). The diffused collagenX is probably in a much lower oligomeric state. Throughout its intra-Golgi trafficking, collagenX, either in punctate or diffuse appearance, was excluded from the interior of Giantin-rings, where co-expressed mCherry-GPI clearly localized (*Figure 5D–F*). Instead, it always resided at the rim, either colocalizing with Giantin or surrounding Giantin-rings as discrete puncta. At later stages, the puncta outside Giantin-rings were probably exocytic carriers targeting to the PM.

We also tested soluble and transmembrane secretory cargos, FM4-moxGFP and GFP-FM4-CD8a, whose aggregation states can be controlled by the small molecule — D/D solubilizer. These two cargos are similar to the ones used previously (*Lavieu et al., 2013*). NRK cells expressing either cargo were treated with D/D solubilizer at 20°C for 2 hr to accumulate and arrest the de-aggregated chimera at the Golgi mini-stack. At 20°C, cells were subsequently subjected to 2 hr of incubation in the presence or absence of D/D solubilizer to either de-aggregate or aggregate the cargo respectively (nocodazole was in the system throughout the procedure). Our previous work has established that secretory cargos such as VSVG are mostly arrested at the medial Golgi under 20°C treatment (*Tie et al., 2016b*). In some experiments, 10 min warm up at 37°C was conducted before imaging. Using this protocol, the re-aggregated GFP-FM4-CD8a and FM4-moxGFP Golgi puncta upon D/D washout were estimated to have 830 ± 30 (n = 184) and 660 ± 50 (n = 127) copies, respectively (*Figure 5—figure supplement 3C,D*). We observed that, when in the de-aggregated state, both soluble and membrane FM4-chimeras localized to the interior of Giantin-rings (*Figure 5G*; *Figure 5—figure supplement 3E*). Intriguingly, once aggregated, they partitioned to the rim as discrete puncta. Therefore, our light microscopic data indicated that large cargos preferentially partition to the cisternal rim, possibly due to their bulky sizes, while conventional or small cargos tend to locate to the interior.

## Discussion

It poses a great challenge to investigate the structure and organization of the Golgi complex by the light microscopy. We established a method to identify the cisternal rim and interior by taking advantage of rim-localized Golgi markers. In addition to quantitative axial localization using the LQ (*Tie et al., 2016b*), we further showed the advantage of nocodazole-induced Golgi mini-stacks in elucidating the molecular organization of the Golgi complex. We analyzed dozens of Golgi residents representing diverse families of proteins for their lateral localizations. The distribution of enzymes is

restricted to the interior of the medial and *trans*-cisternae. In contrast, trafficking machinery components appear to complement Golgi enzymes by residing at the rim of medial and *trans*-cisternae, entire *cis*-Golgi cisternae and *trans*-Golgi/TGN. Previous EM studies on lateral localizations of trafficking machinery components, including COPI (*Orci et al., 1997*), giantin (*Koreishi et al., 2013*), KDEL receptor (*Cosson et al., 2002*; *Martinez-Menárguez et al., 2001*; *Orci et al., 1997*), GS27 (*Cosson et al., 2005*) and GS15 (*Cosson et al., 2005*), Golgi enzymes, including Man1B1 (*Rizzo et al., 2013*), ManII (*Cosson et al., 2002*; *Cosson et al., 2005*; *Martinez-Menárguez et al., 2001*; *Orci et al., 2000*), MGAT1 (*Orci et al., 2000*) and GalT (*Cosson et al., 2005*), and Golgi-transiting cargos including VSVG (*Martinez-Menárguez et al., 2001*; *Mironov et al., 2001*) and soluble aggregated FM4-fusion protein (*Volchuk et al., 2000*), which are summarized and compared in *Supplementary file 1* and *2*, are mostly consistent with our observations. Our qualitative and quantitative data sketch a Golgi mini-stack as spindle-shaped with medial-cisternae possessing a larger diameter than both *cis*- and *trans*-cisternae (*Figure 4J*). Our morphological description of the Golgi mini-stack, such as the spindle shape of the stack and organization of the TGN, bear similarities to the plant Golgi mini-stack observed by electron tomography (*Staehelin and Kang, 2008*), probably due to the lack of microtubule cytoskeleton in plants, which is similar to nocodazole-treated mammalian cells. Our findings suggest the spatial partition of the processing and transport function to the interior and rim of the Golgi stack, as depicted by our model in *Figure 6*.

EM studies have revealed that cisternal rims are dilated with a width of ~100 nm, while their stacked interiors are narrow and tightly spaced with a width of ~20 nm (*Bykov et al., 2017*; *Engel et al., 2015*; *Staehelin and Kang, 2008*). Recently, zipper-like intracisternal and intercisternal

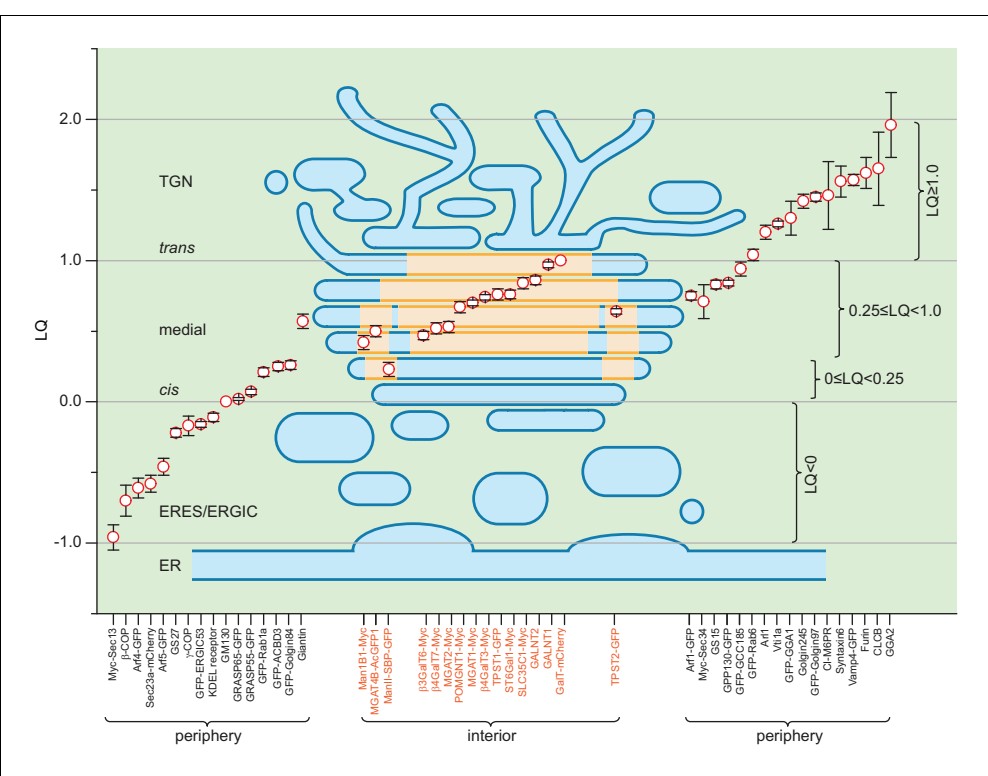

**Figure 6.** A schematic model summarizing the organization of a Golgi mini-stack. LQs of various Golgi residents (see *Table 1*) are overlaid onto a simplified diagram of a Golgi mini-stack together with the ERES and ERGIC. The red circle represents the mean of the LQ with flanking black bars representing the SEM. The cisternal interior, including central disks and inner-rings, is shaded yellow while the periphery of the Golgi mini-stack, including the cisternal rim, is shaded blue. Within the plot, red circles representing Golgi enzymes (labeled orange at the x-axis) are overlaid onto the yellow-shaded interior region, while those of components of the transport machinery (labeled black at the x-axis) are outside the mini-stack to indicate their periphery localization.
DOI: https://doi.org/10.7554/eLife.41301.020

protein arrays have been discovered at interior regions of medial and *trans*-cisternae in green alga through the cryo-electron tomography (*Engel et al., 2015*). It was proposed that these tightly packed protein arrays comprise Golgi enzymes. Our super-resolution and EM data from the Golgi mini-stack provide direct evidence supporting this hypothesis. These enzyme-arrays might organize as an 'enzyme matrix' to 1) stack cisternal membrane, 2) retain Golgi enzymes or accessory proteins and 3) exclude trafficking machinery components by a possible molecular crowding mechanism. Therefore, it seems that, collectively, Golgi enzymes determine and maintain the characteristic structure of the Golgi complex.

Most secretory cargos in higher eukaryotes undergo glycosylation in the Golgi complex. Our finding that the cisternal interior and rim correspond to processing and transport domain, respectively, implies that secretory cargos must access interior domains of different cisternae and then reside there long enough for sequential glycosylation. This is indeed what we observed for conventional cargos, such as GPI-anchored proteins, Furin, E-cadherin, VSVG and secretory GFP. On the other hand, these cargos probably have a sufficiently short residence time in the cisternal rim, in which they are either retrieved and retained by the 'enzyme matrix' to the interior or packed into membrane carriers targeting to the PM at the *trans*-Golgi. It seems that the retention by the 'enzyme matrix' occurs by default and is independent of glycosylation because secretory GFP is preferentially found within the cisternal interior. However, this is not the case for bulky cargos, such as collagenX and aggregated GFP-FM4-CD8a and FM4-moxGFP, which localized only at the rim and were excluded from the interior. These observations suggest that bulky cargos might be incompatible with the crowded molecular environment of the tightly packed 'enzyme matrix' and/or the narrow luminal space at the interior, which can have a width of <10 nm (*Engel et al., 2015*). Rim partitioning of large secretory cargos has previously been noted by EM (*Bonfanti et al., 1998*; *Engel et al., 2015*; *Lavieu et al., 2013*; *Volchuk et al., 2000*). Here, we directly visualized by light microscopy the size-dependent lateral partitioning of secretory cargos within the Golgi stack.

This study did not attempt to resolve different intra-Golgi trafficking models and our discoveries can be explained by both cisternal progression and stable compartment models or their modified variants. Nonetheless, our findings provide important insight into the structure and organization of the Golgi complex.

# Materials and methods

## Key resources table

| Reagent type (species) or resource | Designation | Source or reference | Identifiers | Additional information |
|---|---|---|---|---|
| Cell line (*Homo sapiens*) | HeLa cell | ATCC | ATCC: CCL-2; RRID:CVCL_0030 | |
| Cell line (*Rattus norvegicus*) | Normal rat kidney (NRK) fibroblast cell | ATCC | ATCC: CRL-1570; RRID:CVCL_2144 | |
| Antibody | GM130 C-terminus (mouse monoclonal) | BD Biosciences | BD Biosciences: 610822; RRID:AB_398141 | (1:500) |
| Antibody | Golgin 245 (mouse monoclonal) | BD Biosciences | BD Biosciences: 611280; RRID:AB_398808 | (1:100) |
| Antibody | GGA2 (mouse monoclonal) | BD Biosciences | BD Biosciences: 612612; RRID:AB_399892 | (1:200) |
| Antibody | GS15 (mouse monoclonal) | BD Biosciences | BD Biosciences: 610960; RRID:AB_398273 | (1:250) |

*Continued on next page*

*Continued*

| Reagent type (species) or resource | Designation | Source or reference | Identifiers | Additional information |
|---|---|---|---|---|
| Antibody | GS27 (mouse monoclonal) | BD Biosciences | BD Biosciences: 611034; RRID:AB_398347 | (1:250) |
| Antibody | GS28 (mouse monoclonal) | BD Biosciences | BD Biosciences: 611184; RRID:AB_398718 | (1:250) |
| Antibody | Syntaxin6 (mouse monoclonal) | BD Biosciences | BD Biosciences: 610635; RRID:AB_397965 | (1:250) |
| Antibody | Vti1a (mouse monoclonal) | BD Biosciences | BD Biosciences: 611220; RRID:AB_398752 | (1:500) |
| Antibody | Myc (mouse monoclonal) | Santa cruz biotechnology | Santa cruz: sc-40; RID:AB_627268 | (1:200) |
| Antibody | CLCB (mouse monoclonal) | Santa cruz biotechnology | Santa cruz: sc-376414; RRID:AB_11149726 | (1:200) |
| Antibody | γCOP (mouse monoclonal) | Santa cruz biotechnology | Santa cruz:sc-393977; RRID:AB_2753138 | (1:200) |
| Antibody | Furin (rabbit polyclonal) | Thermo Fisher Scientific | Thermo Fisher Scientific: PA1062; RRID:AB_2105077 | (1:100) |
| Antibody | CI-M6PR (mouse monoclonal) | Thermo Fisher Scientific | Thermo Fisher Scientific: MA1066; RRID:AB_2264554 | (1:200) |
| Antibody | Alexa Fluor 594 conjugated streptavidin | Thermo Fisher Scientific | Thermo Fisher Scientific: S11227; RRID:AB_2313574 | (1:500) |
| Antibody | βCOP (mouse monoclonal) | Sigma-Aldrich | Sigma-Aldrich: G6160; RRID:AB_477023 | (1:200) |
| Antibody | Flag (mouse monoclonal) | Sigma-Aldrich | Sigma-Aldrich: F3165; RRID:AB_259529 | (1:200) |
| Antibody | GM130 N-terminus (rabbit monoclonal) | Abcam | Abcam: ab52649; RRID:AB_880266 | (1:500) |
| Antibody | Giantin N-terminus (rabbit polyclonal) | BioLegend | Biolegend: 924302; RRID:AB_2565451 | (1:1000) |
| Antibody | Giantin C-terminus (rabbit polyclonal) | this paper | | (1:500); rabbit polyclonal; against aa 3131–3201 |
| Antibody | KDEL receptor (mouse monoclonal) | Enzo Life Sciences | Enzo Life Sciences: VAA-PT048D; RRID:AB_1083549 | (1:250) |
| Antibody | GALNT1 | Other | | (1:10); H Clausen lab (University of Copenhagen) |
| Antibody | GALNT2 | Other | | (1:10); H Clausen lab (University of Copenhagen) |
| Antibody | Arl1 (rabbit polyclonal) | PMID: 11792819 | | (1:100) |

*Continued on next page*

*Continued*

| Reagent type (species) or resource | Designation | Source or reference | Identifiers | Additional information |
|---|---|---|---|---|
| Antibody | Golgin97 (rabbit polyclonal) | PMID: 12972563 | | (1:1000) |
| Recombinant DNA reagent | pDMyc-neo-N1 | this paper | | See *Supplementary file 3* |
| Recombinant DNA reagent | pDMyc-Neo | PMID: 12972563 | | |
| Recombinant DNA reagent | pGEB | PMID: 11792819 | | |
| Recombinant DNA reagent | pA2E-N1 | PMID: 27369768 | | |
| Recombinant DNA reagent | pmCherry-C2 | this paper | | See *Supplementary file 3* |
| Recombinant DNA reagent | Streptavidin-His | PMID: 16554831 | RRID:Addgene_20860 | Addgene plasmid #20860 |
| Recombinant DNA reagent | Strep-Ii_VSVG-SBP-EGFP | PMID: 22406856 | RRID:Addgene_65300 | Addgene plasmid #65300 |
| Recombinant DNA reagent | ss-Strep-KDEL_ManII-SBP-GFP | PMID: 22406856 | RRID:Addgene_65252 | Addgene plasmid #65252 |
| Recombinant DNA reagent | ss-Strep-KDEL_ss-SBP-mCherry-GPI | PMID: 22406856 | RRID:Addgene_65295 | Addgene plasmid #65295 |
| Recombinant DNA reagent | TPST1-GFP | PMID: 18522538 | RRID:Addgene_66617 | Addgene plasmid #66617 |
| Recombinant DNA reagent | TPST2-GFP | PMID: 18522538 | RRID:Addgene_66618 | Addgene plasmid #66618 |
| Recombinant DNA reagent | pmScarlet-Giantin-C129 | PMID: 27869816 | RRID:Addgene_85048 | Addgene plasmid #85048 |
| Recombinant DNA reagent | li-Strep_ss-SBP-GFP | this paper | | RUSH reporter of soluble SBP-GFP |
| Recombinant DNA reagent | Strep-Ii_VSVG-SBP-Flag | this paper | | RUSH reporter of VSVG-SBP-Flag |
| Recombinant DNA reagent | ss-Strep-KDEL_ss-SBP-GFP-E-cadherin | PMID: 22406856 | | RUSH reporter of SBP-GFP-E-cadherin; a gift from F. Perez lab (Institut Curie) |
| Recombinant DNA reagent | ss-Strep-KDEL_ss-SBP-GFP-CD8a-Furin | PMID: 26764092 | | RUSH reporter of SBP-GFP-CD8a-Furin |
| Recombinant DNA reagent | ss-Strep-KDEL_ss-SBP-GFP-CD59 | PMID: 26764092 | | RUSH reporter of SBP-GFP-CD59 |
| Recombinant DNA reagent | ss-Strep-KDEL_ss-SBP-GFP-collagenX | Other | | RUSH reporter of SBP-GFP-collagenX; a gift from F Perez lab (Institut Curie) |
| Recombinant DNA reagent | Rab1a-GFP | this paper | | See *Supplementary file 3* |
| Recombinant DNA reagent | Furin-GFP | this paper | | See *Supplementary file 3* |
| Recombinant DNA reagent | Fuin-Myc | this paper | | See *Supplementary file 3* |
| Recombinant DNA reagent | GFP-GCC185 | this paper | | See *Supplementary file 3* |
| Recombinant DNA reagent | GFP-GCC185-mCherry | this paper | | See *Supplementary file 3* |

*Continued on next page*

*Continued*

| Reagent type (species) or resource | Designation | Source or reference | Identifiers | Additional information |
|---|---|---|---|---|
| Recombinant DNA reagent | GFP-ACBD3 | this paper | | See *Supplementary file 3* |
| Recombinant DNA reagent | GFP-Rab6 | this paper | | See *Supplementary file 3* |
| Recombinant DNA reagent | mCherry-Golgin84 | this paper | | See *Supplementary file 3* |
| Recombinant DNA reagent | GFP-GGA1 | this paper | | See *Supplementary file 3* |
| Recombinant DNA reagent | mCherry-GM130 | this paper | | See *Supplementary file 3* |
| Recombinant DNA reagent | Arf1-GFP | PMID: 16890159 | | A gift from FJM van Kuppeveld lab (Utrecht University) |
| Recombinant DNA reagent | Arf4-GFP | Other | | A gift from FJM van Kuppeveld lab (Utrecht University) |
| Recombinant DNA reagent | Arf5-GFP | Other | | A gift from FJM van Kuppeveld lab (Utrecht University) |
| Recombinant DNA reagent | GFP-ERGIC53 | PMID: 15632110 | | A gift from H Hauri lab (University of Basel) |
| Recombinant DNA reagent | GFP-GM130 | PMID: 11781572 | | A gift from M De Matties lab (Telethon Institute of Genetics and Medicine, Italy) |
| Recombinant DNA reagent | GFP-Golgin84 | PMID: 12538640 | | A gift from M Lowe lab (University of Manchester) |
| Recombinant DNA reagent | GFP-Golgin97 | PMID: 11792819 | | A gift from W Hong lab (Institute of Molecular and Cell Biolgoy, Singapore) |
| Recombinant DNA reagent | GPP130-GFP | PMID: 9201717 | | A gift from A Linstedt lab (Carnegie Mellon University) |
| Recombinant DNA reagent | GRASP55-GFP | Other | | A gift from Y Zhuang lab (University of Michigan) |
| Recombinant DNA reagent | GRASP65-GFP | Other | | A gift from Y Zhuang lab (University of Michigan) |
| Recombinant DNA reagent | DMyc-GCC185 | Other | | A gift from W Hong lab (Institute of Molecular and Cell Biolgoy, Singapore) |

*Continued on next page*

Continued

| Reagent type (species) or resource | Designation | Source or reference | Identifiers | Additional information |
|---|---|---|---|---|
| Recombinant DNA reagent | Sec23a-mCherry | Other | | A gift from W Hong lab (Institute of Molecular and Cell Biolgoy, Singapore) |
| Recombinant DNA reagent | Sec31a-GFP | PMID: 10788476 | | A gift from W Hong lab (Institute of Molecular and Cell Biolgoy, Singapore) |
| Recombinant DNA reagent | Vamp4-GFP | PMID: 17327277 | | A gift from W Hong lab (Institute of Molecular and Cell Biolgoy, Singapore) |
| Recombinant DNA reagent | Myc-Sec34 | PMID: 11929878 | | A gift from W Hong lab (Institute of Molecular and Cell Biolgoy, Singapore) |
| Recombinant DNA reagent | Myc-Sec13 | PMID: 22609279 | | A gift from W Hong lab (Institute of Molecular and Cell Biolgoy, Singapore) |
| Recombinant DNA reagent | MGAT1-AcGFP1 | this paper | | See *Supplementary file 3* |
| Recombinant DNA reagent | MGAT2-AcGFP1 | this paper | | See *Supplementary file 3* |
| Recombinant DNA reagent | MGAT4B-AcGFP1 | this paper | | See *Supplementary file 3* |
| Recombinant DNA reagent | ST6Gal1-AcGFP1 | this paper | | See *Supplementary file 3* |
| Recombinant DNA reagent | Man1B1-Myc | this paper | | See *Supplementary file 3* |
| Recombinant DNA reagent | MGAT1-Myc | this paper | | See *Supplementary file 3* |
| Recombinant DNA reagent | MGAT2-Myc | this paper | | See *Supplementary file 3* |
| Recombinant DNA reagent | ST6Gal1-Myc | this paper | | See *Supplementary file 3* |
| Recombinant DNA reagent | β4GalT3-Myc | this paper | | See *Supplementary file 3* |
| Recombinant DNA reagent | GalT-mCherry | PMID: 26764092 | | |
| Recombinant DNA reagent | SLC35C1-Myc | OriGene Technologies Inc. | Cat. No.: RC200101 | |
| Recombinant DNA reagent | β3GalT6-Myc | OriGene Technologies Inc. | Cat. No.: MR204731 | |
| Recombinant DNA reagent | β4GalT7-Myc | OriGene Technologies Inc. | Cat. No.: RC200258 | |
| Recombinant DNA reagent | POMGNT1-Myc | OriGene Technologies Inc. | Cat. No.: RC200176 | |

*Continued on next page*

*Continued*

| Reagent type (species) or resource | Designation | Source or reference | Identifiers | Additional information |
|---|---|---|---|---|
| Recombinant DNA reagent | FM4-moxGFP | this paper | | See *Supplementary file 3* |
| Recombinant DNA reagent | GFP-FM4-CD8a | PMID: 23755362 | | A gift from James Rothman lab (Yale University) |
| Recombinant DNA reagent | GPP130-APEX2-GFP | this paper | | See *Supplementary file 3* |
| Recombinant DNA reagent | MGAT2-APEX2-GFP | this paper | | See *Supplementary file 3* |
| Recombinant DNA reagent | His-Giantin(3131–3201) | this paper | | See *Supplementary file 3* |
| Recombinant DNA reagent | GST-Giantin(3131–3235) | this paper | | See *Supplementary file 3* |
| Recombinant DNA reagent | GFP-Nup133-mut | PMID: 27613095 | | See *Supplementary file 3* |
| Recombinant DNA reagent | shNup133-1 | PMID: 27613095 | | See *Supplementary file 3* |
| Commercial assay or kit | APEX Alexa Fluor 488 Antibody Labeling Kit | Thermo Fisher Scientific | Invitrogen: A10475 | |
| Commercial assay or kit | APEX Alexa Fluor 488 Antibody Labeling Kit | Thermo Fisher Scientific | Invitrogen A10468 | |
| Chemical compound, drug | biotin | IBA | IBA: 21016002 | 40 µM |
| Chemical compound, drug | biotin phenol | Iris Biotech GmbH | Iris Biotech GmbH: LS3500 | 500 µM |
| Chemical compound, drug | nocodazole | Merck | Merck: 487928 | 33 µM |
| Chemical compound, drug | D/D solubilizer | Clontech | Clontech: 635054 | 1 mM |
| Software, algorithm | Fiji | PMID: 22743772 | https://fiji.sc/ | |
| Software, algorithm | Calculation of the LQ | PMID: 26764092; PMID: 28829416 | | |
| Software, algorithm | Gyradius and intensity normalization.ijm | this paper | | To normalize diameters and intensities of en face Golgi mini-stacks |
| Software, algorithm | Golgi mini-stack alignment.ijm | this paper | | To align normalized en face Golgi mini-stacks |
| Software, algorithm | Radial mean intensity profile.ijm | this paper | | To measure radial mean intensity of en face averaged Golgi mini-stacks |

## DNA plasmids

See *Supplementary file 3*.

## Antibodies and small molecules

The following mouse monoclonal antibodies (mAbs) were purchased from BD Biosciences: GM130 C-terminus, Golgin245, GGA2, GS15, GS27, GS28, Syntaxin6 and Vti1a. The following mouse mAbs were from Santa Cruz: Myc, CLCB and γCOP. Rabbit polyclonal antibody (pAb) against Furin, mouse

mAb against CI-M6PR and Alexa Fluor 594 conjugated streptavidin were from Thermo Fisher Scientific. The following antibodies were commercially available from respective vendors: mouse mAb against Flag-tag and βCOP (Sigma-Aldrich), rabbit mAb against the N-terminus of GM130 (Abcam), rabbit pAb against Giantin (BioLegend) and mouse mAb against KDEL receptor (Enzo Life Sciences). Mouse mAbs against GALNT1 and GALNT2 were from H. Clausen. Rabbit pAbs against Arl1 and Golgin97 were previously described (*Lu and Hong, 2003*; *Lu et al., 2001*). The following small molecules were commercially available: biotin (IBA), biotin phenol (Iris Biotech GmbH), nocodazole (Merck) and D/D solubilizer (Clontech).

## Cell lines

HeLa and normal rat kidney fibroblast (NRK) cells were from American Type Culture Collection. Cell were assumed to be authenticated by the supplier. The presence of mycoplasma contamination was monitored by Hoechst 33342 staining.

## Cell culture and transfection

HeLa and NRK cells were cultured in Dulbecco's Modified Eagle's Medium (DMEM) supplemented with 10% fetal bovine serum. Cell transfection was conducted using *Orci et al., 2000* (Invitrogen) according to manufacturer's manual. In live-cell imaging, cells grown on a Φ35 mm glass-bottom Petri-dish (MatTek) were imaged in $CO_2$-independent medium (Invitrogen) supplemented with 4 mM glutamine and 10% fetal bovine serum at 37°C. Unless otherwise indicated, all cells used were HeLa and treated with 33 μM nocodazole to induce the formation of Golgi mini-stacks.

## Production of Giantin C-terminal antibody

It was conducted as previously described (*Madugula and Lu, 2016*; *Mahajan et al., 2013*). Briefly, His-Giantin(3131–3201) was purified in urea from bacteria and used as the antigen to raise the antiserum in rabbits (Genemed Synthesis Inc). Recombinant GST-Giantin(3131–3235) was purified from bacteria and subsequently used to purify the antibody from the anti-serum.

## Super-resolution fluorescence microscopy

The Airyscan super-resolution microscope system (Carl Zeiss) comprises a Zeiss LSM710 confocal microscope equipped with an oil objective lens (alpha Plan-Apochromat 100 ×, 1.46 NA), a motorized stage, a temperature control environment chamber and Airyscan module. Fluorophores were excited by three laser lines with wavelengths of 488, 561 and 640 nm and their respective emission bandwidths were 495–550 nm, 595–620 nm and long pass 645 nm. The microscope system was controlled by ZEN software (Carl Zeiss). Pixel size of images ranged from 40 to 54 nm. For 3D imaging, the z-step of image stacks was 170 nm. Image stacks were subjected to Airyscan processing and maximal intensity projection (MIP) in ZEN software. Image analysis was performed in Fiji (https://imagej.net/Fiji). We exhausted our images for all Golgi mini-stacks that were visually identifiable as either en face or side views.

## En face averaging of golgi mini-stack images and radial mean intensity profile acquisition

En face view images of Giantin-labeled Golgi mini-stacks were averaged in semi-automatic software tools that were developed using macros of Fiji. Mini-stack images were first cropped to square shape and subjected to background subtraction. To quantify the size of the Giantin-ring, we adopted the concept of the gyradius from physics. For pixel i in the Giantin-ring image, assuming that $I_i$ is its intensity and $r_i$ is its distance to the center of fluorescence mass, the gyradius of the Giantin-ring can be calculated as

$$\sqrt{\frac{\sum(I_i \cdot r_i^2)}{\sum I_i}},$$

with all pixels of the image considered. The macro 'gyradius and intensity normalization' (see *Source code 1*) calculates the gyradius of Giantin in a set of multi-channel images and resizes the set of images so that the gyradius of Giantin is 100 pixels. The canvas of the image set is further

expanded to 701 × 701 pixel. Using the macro 'Golgi mini-stack alignment' (see *Source code 2*), Golgi marker images are aligned so that their centers of fluorescence mass are at (350, 350), the center of the image. The en face averaged Golgi mini-stack image is acquired by z-projection of these aligned images. The radial mean intensity profile is acquired using the macro 'Radial mean intensity profile' (see *Source code 3*). The mean intensity of all pixels within a circle around the center of the fluorescence mass is plotted against its radius (ranging from 1 to 350 pixels). The radius of a Golgi marker is defined by the half maximum position of its outer slope of the intensity plot and is normalized by the corresponding radius of Giantin. Detailed steps are described in *Supplementary file 4*.

### Measuring diameters of Giantin-rings

To measure the diameter of a Giantin-ring, a line was first drawn across its center. In the resulting line intensity profile (Fiji), the diameter of the ring was defined as the distance between the two half-maximum-intensity points at outer slopes.

### Immunofluorescence labeling and RUSH cargo trafficking assay

These were conducted as previously described (*Tie et al., 2016b*). By default, tagged-proteins were transiently transfected while non-tagged proteins were native and immuno-stained by their antibodies.

### Fluorescence labeling of APEX2-mediated biotinylation

Nocodazole-treated HeLa cells expressing MGAT2-APEX2-GFP were incubated with 500 μM biotin phenol for 30 min at 37°C. Cells were subsequently transferred to ice and treated with 1 mM $H_2O_2$ for 1 min with brief agitation. After extensive washing with PBS containing 10 mM sodium ascorbate (Sigma-Aldrich), 5 mM Trolox (Sigma-Aldrich), and 10 mM sodium azide (Sigma-Aldrich), cells were fixed and processed for immunofluorescence. Biotinylated proteins were labeled by Alexa Fluor 594 conjugated streptavidin.

### APEX2-EM

EM was performed as previously described (*Ludwig et al., 2016*) with minor modifications. In brief, NRK cells transiently expressing GPP130-APEX2-GFP or MGAT2-APEX2-GFP were fixed with 2% glutaraldehyde in 0.1 M cacodylate buffer pH 7.4 (CB) containing 2 mM $CaCl_2$ for 1 hr on ice, rinsed three times in CB, and incubated in 0.5 mg/ml 3,3'-diaminobenzidine and 0.5 mM $H_2O_2$ in CB for 5 min. Cells were washed several times in CB and post-fixed in 1% osmium tetroxide in CB containing 2 mM $CaCl_2$ supplemented with 1% (w/v) potassium ferricyanide for 1 hr on ice in the dark. Samples were further processed as described previously (*Ludwig et al., 2016*). After image acquisition, only Golgi stacks with long axis >500 nm were analyzed.

### Calculation of the LQ

The LQ of a Golgi protein was acquired as previously described using a conventional wide-field fluorescence microscope (*Tie et al., 2017*; *Tie et al., 2016b*).

### Estimating the stoichiometry of the fluorescence protein aggregate

This was performed using our previously established method (*Tie et al., 2016a*). HeLa cells were co-transfected with shNup133-1 and GFP-Nup133-mut to knock down the endogenous Nup133 and replace it with shRNA-resistant GFP-Nup133-mut. The resulting nuclear pores, which contain ~16 GFP-Nup133-mut (*Tie et al., 2016a*), were used as a fluorescence standard to quantify the copy number of GFP-collagenX, GFP-FM4-CD8a and FM4-moxGFP at Golgi-localized puncta. Identical imaging conditions were used under Airyscan super-resolution microscopy to image Nup133 and the fluorescence protein aggregate puncta. In GFP-Nup133-mut image, a circular region of interest (ROI) that contains a nuclear pore was generated and its total intensity was quantified as $I_{Nup}$ (*Tie et al., 2016a*). The total intensity of a circular ROI containing a Golgi punctum was also similarly acquired as $I_{punctum}$. The copy number of GFP-tagged chimera in the Golgi punctum was therefore calculated as $16 \times I_{punctum}/ I_{Nup}$. To quantify the copy number of FM4-moxGFP, moxGFP was assumed to be 1.47-fold brighter than EGFP (https://www.addgene.org/fluorescent-proteins/

plasmid-backbones/), which is called GFP in this study, and the copy number of FM4-moxGFP in the Golgi punctum was calculated as $10.9 \times I_{punctum}/ I_{Nup}$.

### Fluorescence-conjugation of Giantin antibodies

Alexa Fluor 647 and Alexa Fluor 488 were covalently conjugated onto a commercial (BioLegend) (against the N-terminus) and our homemade (against the C-terminus) rabbit pAb against Giantin, respectively, using APEX antibody labeling kit (Invitrogen) according to the manufacturer's protocol.

## Acknowledgements

We would like to thank A Luini for discussions and the following persons for sharing their reagents with us: H Clausen, M De Matties, D Gadella, H Hauri, W Hong, A Linstedt, M Lowe, F Perez, J Rothman, E Snapp, Z Song, D Stephens, A Ting, F van Kuppeveld and Y Zhuang. This work was supported by the following grants to LL: NMRC/CBRG/007/2012, MOE AcRF Tier1 RG132/15, Tier1 RG35/17, Tier1 RG48/13 and Tier2 MOE2015-T2-2-073. This work was further supported (to SS) by the NTU Institute of Structural Biology (NISB) and a MOE Tier3 grant (MOE2012-T3-1-001).

## Additional information

### Funding

| Funder | Grant reference number | Author |
|---|---|---|
| Ministry of Education - Singapore | Tier3 MOE2012-T3-1-001 | Sara Sandin |
| Ministry of Education - Singapore | Tier1 RG132/15 | Lei Lu |
| Ministry of Education - Singapore | Tier1 RG35/17 | Lei Lu |
| Ministry of Education - Singapore | Tier1 RG48/13 | Lei Lu |
| Ministry of Education - Singapore | Tier2 MOE2015-T2-2-073 | Lei Lu |
| National Medical Research Council | NMRC/CBRG/007/2012 | Lei Lu |

The funders had no role in study design, data collection and interpretation, or the decision to submit the work for publication.

### Author contributions

Hieng Chiong Tie, Data curation, Formal analysis, Investigation, Methodology, Writing—review and editing; Alexander Ludwig, Resources, Formal analysis, Investigation, Visualization, Methodology, Writing—review and editing; Sara Sandin, Resources, Funding acquisition; Lei Lu, Conceptualization, Resources, Data curation, Software, Formal analysis, Supervision, Funding acquisition, Validation, Investigation, Visualization, Methodology, Writing—original draft, Project administration, Writing—review and editing

### Author ORCIDs

Hieng Chiong Tie (iD) http://orcid.org/0000-0003-2738-8685
Alexander Ludwig (iD) http://orcid.org/0000-0002-0696-5298
Lei Lu (iD) http://orcid.org/0000-0002-8192-1471

### Decision letter and Author response

Decision letter https://doi.org/10.7554/eLife.41301.030
Author response https://doi.org/10.7554/eLife.41301.031

# Additional files

## Supplementary files

• Source code 1. Fiji macro 'Gyradius and intensity normalization.ijm'.
DOI: https://doi.org/10.7554/eLife.41301.021

• Source code 2. Fiji macro 'Golgi mini-stack alignment.ijm'.
DOI: https://doi.org/10.7554/eLife.41301.022

• Source code 3. Fiji macro 'Radial mean intensity profile.ijm'.
DOI: https://doi.org/10.7554/eLife.41301.023

• Supplementary file 1. Mouse Genome Informatics (MGI) and Human Genome Organization Gene Nomenclature Committee (HGNC) official full names of glycosylation enzymes used in this study. The official full name of ManII is from MGI while the rest are from HGNC. Except GalT and ManII, all names are official symbols.
DOI: https://doi.org/10.7554/eLife.41301.024

• Supplementary file 2. Review of previous EM literature that directly addressed lateral localizations of Golgi residents and secretory cargos in comparison with this study.
DOI: https://doi.org/10.7554/eLife.41301.025

• Supplementary file 3. The source and cloning method of DNA plasmids used in this study.
DOI: https://doi.org/10.7554/eLife.41301.026

• Supplementary file 4. Protocol for en face averaging and radial mean intensity profile.
DOI: https://doi.org/10.7554/eLife.41301.027

• Transparent reporting form
DOI: https://doi.org/10.7554/eLife.41301.028

## Data availability

All data generated or analysed during this study are included in the manuscript and supporting files.

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
