## [Decision Letter]

Thank you for submitting your article "The spatial separation of processing and transport functions to the interior and periphery of the Golgi stack" for consideration by *eLife*. Your article has been reviewed by three peer reviewers, including Suzanne R Pfeffer as the Reviewing Editor and Reviewer #1, and the evaluation has been overseen by Anna Akhmanova as the Senior Editor. The following individuals involved in review of your submission have agreed to reveal their identity: Frederic A Bard (Reviewer #2); Alberto Luini (Reviewer #3).

The reviewers have discussed the reviews with one another and the Reviewing Editor has drafted this decision to help you prepare a revised submission.

This is a high quality description of the spatial distribution of Golgi enzymes and trafficking machinery in nocodazole-induced Golgi ministacks using super resolution light microscopy and the RUSH system to monitor transit through the compartment. The authors find that glycosyltransferases appear to reside in the internal portion of the cisternae while trafficking machinery can be found at the rims. The work will be a valuable resource to readers of *eLife* if it was made more scholarly and quantification is added. We suggest that the revision be submitted in the category of a Tools and Resources article.

Most of the suggested edits are textual but important. Two experiments and some additional quantifications are required for revision.

1) The authors use different conditions to study the distribution of enzymes and of cargo proteins. They examine the former in unperturbed cells, and the latter during a traffic wave. However, intense traffic could alter the distribution of the enzymes. Please examine the distribution of at least a couple of enzymes during a traffic pulse. The results might be very interesting and will add something novel to the story.

2) The EM is of low quality and needs quantification. All EM localizations need to be quantified with a suitable N value by some metric to convince the reader of enzyme distributions across the stacks. In addition, the authors conclude that "the total amount of GnT2 in these vesicles was probably much less than that in the interior since we did not find APEX2-signal outside Giantin-rings by fluorescence imaging of Golgi mini-stacks". APEX is non-linear in terms of the signal it generates and so one must be careful about concluding too much from an amplified signal. The fact that the authors did not find the APEX2-signal of GnT2 outside giantin-rings by fluorescence does not mean that GnT2 is less in vesicles than the interior. Only immunogold can tell you absolute amounts of protein, but we are not asking you to carry out immuno EM for this study.

Other comments:

1) In Figure 4A and 4D, the authors show the peripheral and central distribution of small and bulky cargos during the intra Golgi-transport. The time points they study are not appropriate.

It has been extensively shown that a secretory cargo crosses the Golgi within 12-15 minutes. Their synchronization protocol does not allow one to understand what stage of intra-Golgi traffic one is observing. It might be more productive to use the 15 degree block protocol and follow cargo transport along the cis to trans axis in the next 15 min.

2) The authors conclude that "large cargos preferentially partition to the cisternal rim, possibly due to their bulky sizes, while conventional or small cargos tend to locate to the interior". A problem here is that transport is synchronized at 20 °C, which is well known to arrest the secretory proteins mainly in the TGN. The TGN is morphologically and functionally different from the Golgi cisternae, and when cargo reaches the TGN, the intra Golgi transport is complete. This experiment is confusing and needs to be described with greater care.

3) Many of the proteins they study are tagged and overexpressed. This is probably necessary for many enzymes, whose endogenous levels are very low. However, overexpression might alter their distribution. This cannot really be fixed but it should be stated clearly and discussed – Orci demonstrated this in previous EM studies.

4) For the Golgi images for the various markers, the line scans reflect individual cells. It would be important to include quantitation of the number of cells showing a particular staining pattern, in addition to selected images. Figure 3G shows a summary but we could not find any values of N for the experiments. Perhaps for example markers, the localizations of 30 cells could be overlaid or a metric used to distinguish donut morphology from spheres? More quantitation is important.

5) The authors need to be more scholarly about previous immuno EM localizations of various Golgi enzymes and trafficking machinery in relation to their own story. Quantitative work of Orci and Klumperman for example, need to be referred to in a table in comparison with the present data. There is also lots of published information regarding glycosyltransferase localization at the EM level. Subsection “Glycosylation enzymes reside at the interior of a Golgi stack”, first paragraph, this was in plants, which build entirely different glycans. Please use mammalian cell references for the precise enzymes studied – and note in the text where this method confirms.

6) Subsection “trans-Golgi and TGN proteins (LQ ≥ 1.0)”. In most cell types, CI-M6PR (and possibly Furin) is in perinuclear late endosomes next to the Golgi, which would explain lack of co-localization with TGN markers. The clathrin stain has been shown previously to be in a distinct TGN microdomain (Brown et al., 2011). Please correct the text.

7) In the Discussion section, they suggest that the enzymes might be a major component of the protein intracisternal matrix that has been previously visualized by other authors by EM. But this is very unlikely to be the case, as Golgi enzymes are expressed in very low copy numbers.

They venture into speculation about the transport mechanism within the Golgi. It is obvious that the distributions at steady state of cargo proteins and enzymes does not allow them to distinguish between transport models, and the authors actually state that they do not want to speculate on their observations in this regard. In the previous paragraph, however, they suggest that the observed lateral distributions can be linked to anterograde vesicular movement of the cargo proteins which, once deposited in the cisterna would reach the central domain of the cisterna to be glycosylated by enzymes residing in that region. This is an interpretation in favor of the anterograde vesicular transport model. Here the authors draw conclusions that are not based on data and at the same time contradict themselves. Please adhere to a simpler unbiased logic in the Discussion.

8) It seems that the sentence "conventional secretory cargos were observed to transit the cisternal interior before exiting the Golgi at the rim" in the Abstract is more an extrapolation from the data than an observed phenomenon. Since it bears on the debate of cisternal maturation versus vesicular transport of cargo, please modify the text.

9) It would improve greatly the accessibility of the paper to have a summary model in the form of a 3D graphic. It would also be interesting (although maybe not for this paper) to have a simple website document summarizing the sub-cellular localisations of all these Golgi proteins.

10) It should be discussed that most enzymes studied are related to the N-glycosylation pathway. It should also be explained which enzymes are not part of this pathway. Indeed, most glycosylation enzymes are involved in specific glycosylation pathways and different pathways may have slightly different enzymatic distributions within the Golgi.

11) Is there evidence that bulky cargoes have less elaborated N-glycans than small cargoes?

12) COPII components localisation: is this specific for nocodazole induced mini-stacks? It is known indeed that mini-stacks form near ERES. Do the authors think these values would hold for a normal Golgi?

13) The authors propose that some enzymes have a different cisternal distribution (inner ring for some). Is it possible to compare with data obtained using FRET, notably by the group of Sakari Kellokumpu?

---

## [Author Response]

1) The authors use different conditions to study the distribution of enzymes and of cargo proteins. They examine the former in unperturbed cells, and the latter during a traffic wave. However, intense traffic could alter the distribution of the enzymes. Please examine the distribution of at least a couple of enzymes during a traffic pulse. The results might be very interesting and will add something novel to the story.

This is a very interesting suggestion. We studied the effect of the synchronized trafficking wave of RUSH reporter, VSVG-SBP-Flag, on the lateral distribution of ST6Gal1-moxGFP. HeLa cells co-expressing VSVG-SBP-Flag, ST6Gal1-moxGFP and GalT-mCherry were treated with biotin to chase VSVG and en face averaged images were acquired (the average of size and intensity normalized and center aligned en face images; please see the revised Materials and methods). Cisternal rim ROI was defined by the donut formed by the radius of en face averaged ST6Gal1-moxGFP (not subjected to traffic wave) and that of Giantin-ring. The percentage of ST6Gal1 within the interior was quantified at different chase time (min) and is shown in revised Figure 5—figure supplement 2 with the corresponding plot of VSVG-SBP-Flag’s LQ vs. time from the same experiment.

We did not observe any obvious change of the interior localization of Golgi enzymes. However, it is an important control to demonstrate the interior localization of enzymes under traffic wave. We have inserted the below text to the Results section: “The secretory cargo wave does not seem to grossly affect the interior distribution of Golgi enzymes, as evidenced by ST6Gal1 (Figure 5—figure supplement 2). […] A more systematic investigation is required to resolve this discrepancy.”

2) The EM is of low quality and needs quantification. All EM localizations need to be quantified with a suitable N value by some metric to convince the reader of enzyme distributions across the stacks. In addition, the authors conclude that "the total amount of GnT2 in these vesicles was probably much less than that in the interior since we did not find APEX2-signal outside Giantin-rings by fluorescence imaging of Golgi mini-stacks". APEX is non-linear in terms of the signal it generates and so one must be careful about concluding too much from an amplified signal. The fact that the authors did not find the APEX2-signal of GnT2 outside giantin-rings by fluorescence does not mean that GnT2 is less in vesicles than the interior. Only immunogold can tell you absolute amounts of protein, but we are not asking you to carry out immuno EM for this study.

We have imaged more Golgi stacks and performed a simple statistical analysis by quantifying the percentage of MGAT2-APEX2 and GPP130-APEX2 positive Golgi stacks that display predominant interior and rim distribution, respectively. From almost 60 Golgi stacks imaged for each construct, we found that, for MGAT2, 93% (n=58 from 14 cells) stacks demonstrated interior localization while, for GPP130, 68% (n=57 from 25 cells) stacks displayed rim localization. We have modified the manuscript to include these data.

We agree that the APEX2-catalyzed reaction is non-linear and deleted the conclusion that “the total amount of GnT2 in these vesicles was probably much less than that in the interior”. Below is the revised corresponding text:

“Noticeably, APEX2-generated electron density was also found in vesicles and budding profiles at the rim (arrow heads in Figure 4I). However, we did not find MGAT2 (Figure 1K and 4C) or MGAT2-APEX2 (Figure 4—figure supplement 1U) signal outside Giantin-rings by fluorescence imaging of Golgi mini-stacks.”

Other comments:1) In Figure 4A and 4D, the authors show the peripheral and central distribution of small and bulky cargos during the intra Golgi-transport. The time points they study are not appropriate.It has been extensively shown that a secretory cargo crosses the Golgi within 12-15 minutes. Their synchronization protocol does not allow one to understand what stage of intra-Golgi traffic one is observing. It might be more productive to use the 15 degree block protocol and follow cargo transport along the cis to trans axis in the next 15 min.

In Figure 4A and 4D of previous version manuscript, our protocol synchronizes the traffic of secretory cargos from the ER. This should not be an issue since we also showed (in Figure 4B, 4E and 4F) plots of LQ vs. time measured from the same experimental data sets. LQ vs. time plots very accurately and numerically indicate the position of secretory cargos within the Golgi stack or the secretory pathway at the time of interest. Take note that LQ is linear and 0.0 and 1.0 correspond to GM130 and GalT-mCherry localizations respectively. Therefore, by combining both en face view imaging and LQ vs. time data (A and B; D and E, F), we are able to tell the lateral distribution (rim or interior) of a cargo when it is at a precise intra-Golgi position along the cis-trans axis of the Golgi mini-stack.

2) The authors conclude that "large cargos preferentially partition to the cisternal rim, possibly due to their bulky sizes, while conventional or small cargos tend to locate to the interior". A problem here is that transport is synchronized at 20 °C, which is well known to arrest the secretory proteins mainly in the TGN. The TGN is morphologically and functionally different from the Golgi cisternae, and when cargo reaches the TGN, the intra Golgi transport is complete. This experiment is confusing and needs to be described with greater care.

In our previous study, we demonstrated that, at 15 and 20°C, the LQ of secretory cargo VSVG-GFP was arrested at −0.06 ± 0.03 (n = 97) and 0.56 ± 0.03 (n= 124), corresponding to the cis*-* and medial-Golgi localization, respectively (Tie et al., 2016). Our data indicated that at least a significant pool of a small secretory cargo does not reach the trans-Golgi or TGN at 20 °C. The LQ value of 0.56 for VSVG-GFP is likely resulted from the almost even distribution from the cis to trans-Golgi cisternae at 20 °C. Therefore, there should be still intra-Golgi transport after warming up the system to 37 °C.

We added the following sentence to give this background information: “Our previous work has established that secretory cargos such as VSVG are mostly arrested at the medial Golgi under 20 °C treatment (Tie et al., 2016)”.

3) Many of the proteins they study are tagged and overexpressed. This is probably necessary for many enzymes, whose endogenous levels are very low. However, overexpression might alter their distribution. This cannot really be fixed but it should be stated clearly and discussed – Orci demonstrated this in previous EM studies.

The problem of the overexpression on a protein’s subcellular localization is a general and valid concern for all studies. It certainly applies to our study. We think that this reviewer refers to the following paper: Dynamic transport of SNARE proteins in the Golgi apparatus by Cosson et al. (2005), where the authors studied the effect of overexpression of SNAREs on its localization in comparison to endogenous ones. We have modified our text in Results by adding the below sentences: “Due to the lack of reagents to detect endogenous proteins, many residents were detected by the overexpression of their tagged fusions (Table 1). Caution must be taken in the interpretation of our data as it has been documented that overexpression can change both the axial and lateral localization of Golgi residents (Cosson et al., 2005).”.

4) For the Golgi images for the various markers, the line scans reflect individual cells. It would be important to include quantitation of the number of cells showing a particular staining pattern, in addition to selected images. Figure 3G shows a summary but we could not find any values of N for the experiments. Perhaps for example markers, the localizations of 30 cells could be overlaid or a metric used to distinguish donut morphology from spheres? More quantitation is important.

We have developed a software tool to average en face view images of Golgi mini-stacks. Golgi mini-stacks were first co-stained for Giantin and a testing Golgi marker. En face view images of a Golgi marker were subsequently size-normalized by radii of their corresponding Giantin-rings, intensity-normalized to 200 million and aligned according to their centers of fluorescence mass. Only Golgi residents localizing to the stack (LQ in between 0 and 1) were averaged. The en face averaging provides a powerful tool to survey a large number of mini-stack images and extract information such as radius and radial mean intensity profile. The radius of a Golgi marker were normalized by that of Giantin (normalized radius) and plotted against the Golgi marker’s LQ. In revamped Figure 4J (previously Figure 3G), the normalized lateral distribution vs. LQ plot was replaced by normalized radius vs. LQ. The number of mini-stack images used for averaging is indicated by n in Figure 4J. Through en face averaging, we further found that TPST2-GFP displays inner ring appearance. Figures have been modified to include these averaged images and corresponding radial mean intensity profiles.

5) The authors need to be more scholarly about previous immuno EM localizations of various Golgi enzymes and trafficking machinery in relation to their own story. Quantitative work of Orci and Klumperman for example, need to be referred to in a table in comparison with the present data.

EM studies on the Golgi mostly concerned the cis-trans or axial localizations. As suggested by this reviewer, we have examined past quantitative EM studies, especially those from Orci, Luini and Klumperman labs, on the lateral distributions of Golgi residents or Golgi transiting secretory cargos. It is difficult to compare EM results with our light microscopy data since definitions of interior and rim are likely to differ. Our attempt of such comparison is now summarized in Appendix 1.

There is also lots of published information regarding glycosyltransferase localization at the EM level. Subsection “Glycosylation enzymes reside at the interior of a Golgi stack”, first paragraph, this was in plants, which build entirely different glycans. Please use mammalian cell references for the precise enzymes studied – and note in the text where this method confirms.

As suggested, we have added the following sentences to give localization examples of mammalian Golgi enzymes involved in the modification of N-glycan. “This observation is consistent with previous EM studies. […] Second, it has been documented that the sub-Golgi localization of enzymes can be cell-type dependent (Velasco et al., 1993).”

6) Subsection “trans-Golgi and TGN proteins (LQ ≥ 1.0)”. In most cell types, CI-M6PR (and possibly Furin) is in perinuclear late endosomes next to the Golgi, which would explain lack of co-localization with TGN markers. The clathrin stain has been shown previously to be in a distinct TGN microdomain (Brown et al., 2011). Please correct the text.

In native cells (without nocodazole treatment), a significant pool of CI-M6PR localizes to the perinuclear late endosome. Therefore, there are CI-M6PR positive punctate structures near the TGN but they neither co-localize with the TGN nor do they represent carriers or budding profiles directly derived from the TGN, as commented by this reviewer. However, in cells treated with nocodazole, CI-M6PR positive late endosomes are no longer perinuclear and, instead, they localize throughout the cytosol. Therefore, except by chances, late endosomes are not commonly found near Golgi mini-stacks and a majority CI-M6PR punctate structures should be membrane profiles derived from the TGN. Hence, we think that the distinct localization pattern of CI-M6PR can be explained by their localization at the different domain of the TGN from other markers. We have cited the suggested JCB paper by Brown et al. for the microdomain of TGN and Ladinsky et al. (1999) for clathrin microdomains, which are projected away from the Golgi stack.

7) In the Discussion section, they suggest that the enzymes might be a major component of the protein intracisternal matrix that has been previously visualized by other authors by EM. But this is very unlikely to be the case, as Golgi enzymes are expressed in very low copy numbers.They venture into speculation about the transport mechanism within the Golgi. It is obvious that the distributions at steady state of cargo proteins and enzymes does not allow them to distinguish between transport models, and the authors actually state that they do not want to speculate on their observations in this regard. In the previous paragraph, however, they suggest that the observed lateral distributions can be linked to anterograde vesicular movement of the cargo proteins which, once deposited in the cisterna would reach the central domain of the cisterna to be glycosylated by enzymes residing in that region. This is an interpretation in favor of the anterograde vesicular transport model. Here the authors draw conclusions that are not based on data and at the same time contradict themselves. Please adhere to a simpler unbiased logic in the Discussion.

We are open to all possible models of intra-Golgi trafficking. Thanks for pointing out this discrepancy. We have modified the corresponding text: “On the other hand, these cargos probably have a sufficiently short residence time in the cisternal rim, in which they are either retrieved and retained by the “enzyme matrix” to the interior or packed into membrane carriers targeting to the PM at the trans-Golgi”.

8) It seems that the sentence "conventional secretory cargos were observed to transit the cisternal interior before exiting the Golgi at the rim" in the Abstract is more an extrapolation from the data than an observed phenomenon. Since it bears on the debate of cisternal maturation versus vesicular transport of cargo, please modify the text.

We have modified it to: “conventional secretory cargos appeared at the cisternal interior during their intra-Golgi trafficking and transiently localized to the cisternal rim before exiting the Golgi.”

9) It would improve greatly the accessibility of the paper to have a summary model in the form of a 3D graphic. It would also be interesting (although maybe not for this paper) to have a simple website document summarizing the sub-cellular localisations of all these Golgi proteins.

We have provided a 2D schematic model to summarize the key findings in Figure 6. We will map more sub-Golgi localizations or LQs of Golgi residents and publish these data in certain form in the near future.

10) It should be discussed that most enzymes studied are related to the N-glycosylation pathway. It should also be explained which enzymes are not part of this pathway. Indeed, most glycosylation enzymes are involved in specific glycosylation pathways and different pathways may have slightly different enzymatic distributions within the Golgi.

As suggested, the following text was added to Results to categorize enzymes in our study: “We studied components of Golgi post-translational modification machinery (Table 1; Supplementary file 1), including a GDP-fucose transporter, SLC35C1 (Lubke et al., 2001), and more than a dozen enzymes involved in N-glycosylation (ManIB1, MGAT1, ManII, MGAT2, GalT, SialT and MGAT4B), O-glycosylation (GalNT1, GalNT2 and POMGNT1), poly-N-acetyllactosamine synthesis (β4GalT3), glycosaminoglycan synthesis (β3GalT6 and β4GalT7) and sulfation (TPST1 and 2)”. Therefore, in addition to N-glycosylation enzymes, enzymes involved in other glycosylation and post-translational modifications were also studied.

11) Is there evidence that bulky cargoes have less elaborated N-glycans than small cargoes?

This is an interesting question that we also wanted to address. We reported in this manuscript that bulky cargoes transit through the rim of Golgi cisternae while Golgi enzymes localize to the interior. According to our knowledge, it is currently unclear whether bulky cargoes have less elaborated N- or O-glycosylation than small cargos. We think that the rim partition probably doesn’t abolish the Golgi-type N-glycosylation modification of large secretory cargos. This is because: (1) ManI, MGAT4B and ManII can localize to the rim; (2) minor amount of enzymes could transiently pass through the rim; and (3) the low amount of glycosylation enzymes could be sufficient to modify the low copy numbers of the large secretory cargos in the rim.

As the best known large secretory cargo, collagen has been noted to be extensively O-glycosylated by galactosyl and glucosyl transferases. Interestingly, different from the O-glycosylation at the Golgi, the O-glycosylation of collagen takes place in the ER before its triple helical structure is assembled (Kadler, 1994). Collagen galactosyltransferase contains RDEL ER localization-motif (Schegg et al., 2009) and has been shown to localize to the ER (Liefhebber et al., 2010). However, some collagens, such as Col4a1 and Col4a2, have single N-glycosylation (Basak et al., 2016), suggesting that collagen might complete the N-glycan modification in the Golgi. Another large secretory cargo, high molecular weight adiponectin, similarly has N-glycan in addition to extensive galactosyl and glucosyl O-glycosylation modifications at the ER.

It is probably difficult to design experiments to address this question since the glycosylation process is usually unique for each protein. Hence, it would be a good strategy if the glycosylation of the same protein can be compared between its monomer and aggregated form. One experiment that we can think of is the small molecule inducible aggregation cargos, such as FM4, engineered with artificial glycosylation motifs. However, for a tight aggregate, it is likely that only surface-exposed glycosylation sites can be accessed and modified by enzymes, resulting in a much lower glycosylation efficiency than monomers and thus obscuring result interpretation.

Therefore, although this is a very interesting question that would be a good follow up study, no conclusion can be made at this stage. We therefore did not discuss it in the revised manuscript.

12) COPII components localisation: is this specific for nocodazole induced mini-stacks? It is known indeed that mini-stacks form near ERES. Do the authors think these values would hold for a normal Golgi?

The LQs of COPII components only apply to the nocodazole-induced Golgi mini-stacks. In native cells that are not treated with nocodazole, COPII and ERES are not physically coupled to the Golgi stacks and their distances can vary greatly. However, we think that LQ values still provide important information for the logical organization of the ERES, ERGIC and Golgi.

13) The authors propose that some enzymes have a different cisternal distribution (inner ring for some). Is it possible to compare with data obtained using FRET, notably by the group of Sakari Kellokumpu?

We found that Man1B1, ManII and MGAT4B appear as inner-rings concentric to corresponding Giantin-rings while other N-glycosylation enzymes localize to the central disk. FRET studies from Kellokumpu lab reported that, among Golgi N-glycosylation enzymes they tested, including MGAT1, MGAT2, GalT and ST6Gal1, only MGAT1/2 and GalT/ ST6Gal1 were found to assemble as heteromeric complexes (Hassinen et al., 2010). Our lateral localization data are consistent with the report from Kellokumpu lab as both MGAT1/2 and GalT/ ST6Gal1 complexes localize to the central disk and the heteromeric interaction might contribute to such interior localization.

References:

Basak T, Vega-Montoto L, Zimmerman LJ, Tabb DL, Hudson BG, Vanacore RM. (2016) Comprehensive Characterization of Glycosylation and Hydroxylation of Basement Membrane Collagen IV by High-Resolution Mass Spectrometry. J Proteome Res. 15(1):245-58.

Hassinen A, Rivinoja A, Kauppila A, Kellokumpu S. (2010) Golgi N-glycosyltransferases form both homo- and heterodimeric enzyme complexes in live cells. J. Biol. Chem. 285(23):17771–17777. doi: 10.1074/jbc.M110.103184

Liefhebber JM, Punt S, Spaan WJ, van Leeuwen HC. (2010) The human collagen beta(1- O)galactosyltransferase, GLT25D1, is a soluble endoplasmic reticulum localized protein. BMC Cell Biol. 11:33.

Kadler K. (1994) Extracellular matrix. 1: fibril-forming collagens. Protein Profile, 1(5):519-638. Review.

Schegg B, Hülsmeier AJ, Rutschmann C, Maag C, Hennet T. (2009) Core glycosylation of collagen is initiated by two beta(1-O)galactosyltransferases. Mol Cell Biol. 29(4):943-52.